

# Definitive evidence of the Mediterranean Outflow heterogeneity. Part 1: at the Strait of Gibraltar entrance

Claude Millot

Les Katikias, 83150 Bandol, France

*Correspondence to:* Claude Millot (ailesetiles@gmail.com)

**Abstract.** All most recent papers about the Mediterranean Outflow (MO) assume that it is homogeneous at least in the western side of the Strait of Gibraltar and that its splitting into veins in the Atlantic Ocean is due to

bathymetric effects at the Strait exit while we demonstrate that proofs about the MO heterogeneity within the whole Strait have been available since the mid 1980's at least. We focus herein on data collected at the Strait entrance in 1985 (Part 1), before analyzing 1985-1986 data within the Strait (Part 2) and data collected during the MO-2009 experiment at the Strait exit (Part 3). Having demonstrated that the MO is markedly heterogeneous from the Strait entrance to the Strait exit, our three papers demonstrate that the splitting into veins is essentially a

direct consequence of the sea functioning. Indeed, veins have hydrological characteristics mainly dependent on those of the intermediate and deep Mediterranean Waters (MWs) formed in both the eastern and the western basins of the Sea, as well as on those of the Atlantic Waters (AWs) that mixed in the Strait with these MWs, the bathymetry at and downstream from the Strait exit playing a negligible role.

Herein, we demonstrate that four-five MWs can be clearly identified at the Strait entrance, as we previously

hypothesized, moreover forming relatively thick and homogeneous superimposed layers that are much more individualized than anywhere else in the Sea. For the first time ever, we provide numerous examples of density instabilities in all these layers that clearly illustrate the processes leading to such an increased stratification at the Strait entrance. So as to motivate theoretical analyses and numerical simulations that appear to be of dramatic interest, we hypothesize that the isopycnals slope (of a few %) across the Strait within the MO itself, that is a

direct consequence of both the Coriolis effect and the different outflowing or overflowing velocities of the MWs, could be the main mechanism responsible for such a layering of the MO.

**Keywords:** Strait of Gibraltar, Mediterranean Outflow, heterogeneity, Mediterranean Waters



**Contents**



# 1 Introduction


The Mediterranean Sea is a machine which, thanks to evaporation exceeding precipitation and river runoff (E>P+R) over and all around the Sea, transforms waters from the Atlantic Ocean (AWs) inflowing at the surface through the Strait of Gibraltar into a set of intermediate and deep Mediterranean Waters (MWs) that will finally form the Mediterranean Outflow (MO). This transformation results from dense water formation processes

occurring, either in the open sea or on continental shelves, during wintertime in the north of several sub-basins of both the eastern and the western basins of the Sea that are linked by the Channel of Sicily. From the Strait exit at least, the MO is then split into veins that spread over a density range of ~0.5 kg.m$^{-3}$ and reach equilibrium depths of 400-1600 m, first along the Iberian continental slope and then in the whole northern Ocean. Even though the MO has been intensively studied from the point of view of strait dynamics (maximal vs. sub-maximal regimes,

inflowing vs. outflowing amounts, tidal internal waves and currents, bottom friction and associated turbulent processes, etc.), very few studies have focused on its hydrological characteristics q (potential temperature), S (salinity) and s$_q$ (potential density anomaly) classically analyzed with q-S diagrams. A large percentage of old (before 2000) papers have considered that the spatial and temporal differences in the MO hydrological characteristics within the Strait were nothing else than natural variability and have postulated the overall

homogeneity of the MO from the Strait entrance (near 5°45'W), claiming that its splitting into veins was then due to bathymetric effects at the Strait exit (near 6°20'W).

With all our papers about the circulation of the MWs within the Sea (up to Millot and Taupier-Letage, 2005a) and the series of papers (since Millot et al., 2006) we have dedicated to the Strait where we claim that several

types of MWs can be continuously identified, we have thus generated an actual and major controversy about the hydrological homogeneity vs. heterogeneity of the MO. We address this controversy with a series of three papers dealing with the Strait entrance (Part 1), the Strait itself (Part 2) and the Strait exit (Part 3) mainly because of the relatively large amount of CTD (Conductivity-Temperature-Depth) data that we either analyze for the first time or re-analyze, be these data relatively old (mid 1980's) for Parts 1 and 2 or recent (late 2000's) for Part 3. Even

though the MO characteristics and dynamics are driven by processes that are different from place to place, this controversy is an overall problem that cannot be truncated, furthermore the most recent papers (2017) now hypothesize that the MO is heterogeneous at the Strait entrance before becoming homogeneous within the Strait itself and then being split into veins just from the Strait exit. However, introducing this controversy, which will be done once herein (Sect. 2), is not an easy task.


The main reason is that the postulate about the MO homogeneity has evolved since, initially about the MO from the Strait entrance, it is now, thanks to our papers, only within the Strait itself; in any case, a splitting of a homogeneous MO due to local interactions with the bathymetry would be a somehow simple process. On the other side, very few old papers have noticed incoherency between such a process and some of their own data

without providing sound explanations; and our series of papers, even though they present clear and coherent arguments for an overall heterogeneity of the MO all along its transit through the Strait, invoke processes that are



relatively complex. Therefore, and just because neither a single in situ experiment nor a single theoretical analysis have ever been dedicated to specify the homogeneity vs. heterogeneity of the MO within the Strait, the actual controversy has to be introduced from a selection of the available papers.


Now, presenting our arguments in such a selection would have provided the reader with a complete but tremendously complicated overview that, we think, is unnecessary since we are now able to clearly demonstrate, with the analysis of a single q-S diagram (Sect. 3), that the MO is definitively heterogeneous within the Strait itself (at 6°05'W and upstream). With such a basic evidence in mind, the reader is thus proposed, all along our

trilogy, to make his/her own point of view about the characteristics of the MO heterogeneity, our personal results and analyzes being only proposed as guidelines in the various Discussion sections.

We then analyze (Sect. 4) a 1985 one-day yo-yo time series consisting in 49 CTD profiles near 35°55'N-5°43'W, which is just east of the Camarinal sills (5°45'W), hence at the Strait entrance. This data set clearly evidences

three groups in the largest densities range and we argue for considering four-five groups in total. We first consider general characteristics that allow the analysis of a contemporary and more classical north-south CTD transect across the Alboran sub-basin at 5°40'W. Having validated our approach with the profiles from the transect, we then analyze in details (Sect. 5) all profiles from the yo-yo time series. We focus on the superimposed homogeneous layers observed on all profiles from both the time series and the transect that give

them a marked step-like structure while vertical density profiles everywhere in the interior of the Sea are much smoother. We show, in particular, some types of θ-S diagrams we personally never saw before and that clearly account for tremendous mixing processes able to transform smooth density profiles into step-like ones. Overall, we show that the MWs at the Strait entrance, which will obviously form the MO, are markedly and coherently heterogeneous over a density range of several 0.01 kg.m$^{-3}$ that is one order of magnitude lower than the range in

the Ocean.

We finally discuss (Sect. 6) some generally forgotten evidences and we specify the key points that must be addressed to clearly understand how a set of MWs smoothly stratified on the vertical within the Sea first come to be markedly stratified on the vertical at the Strait entrance, in both places at a relatively low scale and over a

relatively small range, before forming the MO.

## 2 Background

First note that the two first sentences of the Introduction illustrate the specific use of the terms Ocean, Sea, basin, sub-basin, Strait and Channel we make (since Millot and Taupier-Letage, 2005a). Our major aim is to specify the use of the terms "basin" vs. "sea" (Within a "Western Mediterranean Sea", why dealing with an "Algerian Basin" and a "Tyrrhenian Sea"?), and even avoid having names such as "the Alboran Sea" and the "Alboran



Basin" both in a single abstract. We thus aim at writing texts as simple, clear and logical as possible, furthermore
both the generic terms and the proper names can possibly be omitted.

Now, providing some background about the major homogeneous vs. heterogeneous controversy is also made
difficult by other minor controversies about what are the MWs possibly evidenced at the Strait entrance, which
needs presenting these MWs, and what are the MWs effectively evidenced at the Strait entrance and possibly
downstream, which cannot be definitive due to the lack of convenient data sets. The lightest of the MWs
possibly encountered at the Strait entrance is WIW, the unique intermediate water (IW) from the western basin
(Western IW; CIESM group, 2001); WIW is sometimes improperly named Winter IW and, as specified in Millot
(2013), it is often forgotten that WIW was first identified in the Strait by Gascard and Richez (1985). Below, one
encounters the set of IWs from the eastern basin that, instead of being logically named EIW (Eastern IW), is
improperly and generally given the name of LIW (Levantine IW) that is only the name of one of its components
(Millot, 2013, 2014b); to be noticed is that we also specified in these papers the misunderstanding we generally
have of the mixing processes that lead to the formation, hence to the definition, of such IWs. Contrary to what
could have been reported (e.g. Garcia-Lafuente et al., 2017), the acronym TDW was first proposed by Millot
(1999) for the set of deep MWs (DWs) overflowing from the eastern basin (that come to be only dense in the
western one), just because these waters have been, up to very recently in some studies and still by a majority of
them, ignored in the western basin; arguments were given by Millot (2009) for differentiating an upper and a
lower parts of this water mass that we now coherently propose to name EDW (Eastern DW). Finally, the densest
MW is the unique one originated from the western basin that should be named WDW (Western DW) instead of
WMDW (since the M stands for Mediterranean).

Even though additional information is provided in Sect.6, we can already specify that we have claimed for the
occurrence and possible identification at the Strait entrance of IWs and DWs from both the western and the
eastern basins since Millot (1999), and for the separation of EDW in two, hence giving a total of five different
MWs, since Millot (2009). It is noteworthy that most studies have assumed, up to Garcia-Lafuente et al. (2015),
that LIW and WMDW where the only components of the MO, and even that "LIW is thought to contribute the
bulk of the outflow" (Garcia-Lafuente et al., 2009); in this respect, the fact that the Fig.1 of Garcia-Lafuente et
al. (2017) only displays LIW and WMDW is illustrative. And even if recent studies (Naranjo et al., 2015) specify
their understanding is now "... in good agreement with the previous study of Millot (2014b)", this agreement is
far from being total. Indeed, and for instance, Garcia-Lafuente et al. (2017) specify that "the Winter IW flows
embedded within LIW" while we clearly differentiate these IWs, and that "TDW results from the mixing of old
WMDW with recent LIW", hence misreporting our own definition and totally forgetting that dense waters are
also formed in the eastern basin that must exit that basin, cross the western basin and finally exit the Sea. Also,
and contrary to e.g. Garcia-Lafuente et al. (2009), we never thought necessary to invoke the uplifting of e.g. the
WMDW (in fact all DWs) through the agency of Bernoulli suction to flow across the Strait (Stommel et al.
(1973), Kinder and Parrilla (1987)).



Whatever the case, we demonstrate herein (Sect. 4) the occurrence of four-five MWs at the Strait entrance in the mid 1980's while we demonstrate in Part 3 that, in 2009, only four MWs could be identified at the Strait exit … without having any objective way neither to identify this or that MW nor to say which of them was possibly

missing. We have also shown (Millot et al., 2006) that MWs at the Camarinal sills in the early 2000's were warmer (~0.3 °C) and saltier (~0.06) than in the early 1980's, probably as a consequence of dramatic changes that occurred in the eastern basin in the mid 1990's (Roether et al., 2007), and we show in Part 3 the tremendous consequences of these long term changes at the Strait exit. Last but not least, and just having a look at the raw data colored in Fig.4a, it is clear that two readers must agree on such a coloring while they can disagree about

which of the MWs is associated with this or that color. This is why, because we want to report about presently available hypotheses and to present our data analyzes in a way as objective as possible, we have preferred continuing (following e.g. Millot 2009, 2014a) to deal, in the data analysis Sect. 4 and 5, with "colored waters" instead of "MWs names". However, to help the readers and navigate them as much as possible while showing there is nothing "mysterious" in the coloring, we provide as Table 1 a correspondence between the colors used in

the figures, the MWs names we propose and the MWs names the community agreed on.

| Colors | MWs Acronyms, names (Part 1) | Common MWs Acronyms (CIESM Group, 2001) | Comments |
|---|---|---|---|
| orange | WIW, Western Intermediate Water | WIW | Sometimes improperly named Winter IW (all MWs are formed in winter) |
| red | EIW, Eastern Intermediate Water | None. Generally named LIW, which is not discriminant enough | LIW is only one (as AIW and CIW) component of EIW in the western basin |
| pink | Upper part of EDW, Eastern Deep Water | TDW (since Millot, 1999, but with D for dense and not deep) | Generally ignored |
| violet | Lower part of EDW, Eastern Deep Water | TDW (since Millot, 1999, but with D for dense and not deep) | Differentiation upper vs. lower only made by us |
| blue | WDW, Western Deep Water | WMDW | The M for Mediterranean is useless |

Table 1. Correspondence between the colors in the figures herein, the acronyms and names we propose (essentially, an intermediate and a deep waters from both the eastern and the western basins) as well as those the community proposed in 2001.


Now, for what concerns the Strait exit, actual studies focusing on the Strait entrance and no more considering only two MWs specify "While up to four MWs are spatially distinguishable east of the main sill of Camarinal in the Strait, most of their differentiating characteristics are eroded after flowing over this restrictive topography due to mixing. West of the sill, therefore, speaking of a unique Mediterranean Water seems more appropriate"

(Naranjo et al., 2015). And Garcia-Lafuente et al. (2017) specify that, even though IWs (resp. DWs) are found in the north (resp. south) of the Alboran just before the Camarinal sills (for reasons completely different from those




we invoke), "the severe mixing and dissipation that takes place … downstream … blurs this spatial pattern and tends to form a rather mixed outflow … in which the MWs are barely distinguishable". Consequently, and even if such an assertion is clearly invalidated herein and by our own overall analysis, it is not surprising that studies

at the Strait exit, furthermore since mainly focusing on pure strait dynamics, assume a homogeneous MO there. Since most recent papers about the Strait exit (e.g. Peliz et al. (2007, 2009), Gasser et al. (2011, 2017), Nash et al. (2012)) do not detail the MO hydrological characteristics and make major reference to the relatively old Baringer and Price (1997a,b), these classical oceanography papers deserve to be synthesized.

Baringer and Price (1997a mainly), hence all other papers referring to them as well, consider that "the MO begins at the Strait exit ... having a very narrow range of $\theta$-S-$\sigma_\theta$ properties"; however, they only specify S$\geq$38.4 and $\sigma_\theta \geq$28.95 kg.m$^{-3}$ there, as well as $\theta$=12.9°C-S=38.45 near Camarinal sills. Whatever, the MO is said "to be composed of Pure Mediterranean Water ... that can be seen as a nearly uniform water that fills the deep Mediterranean Sea" and "to mix only with the North Atlantic Central Water (NACW)", hence forgetting to

mention the dramatic role always played in most of the Strait by the Surface Atlantic Water (SAW), while the "bottom density mixed layer within the Strait could be ascribed to the initial condition". From a dynamical point of view, the MO is said to be "diverted by local bathymetric features at the Strait exit" while its descent is said to be "very asymmetric, with the southern/offshore edge descending ~1000 m and the northern/onshore edge descending only ~200 m". The occurrence of two cores or veins (we should nowadays deal with the two (out of

four) major cores known as the Upper and the Lower Cores, see below) is said to be "a consequence of topographic steering, which split the outflow into two partially connected branches". Note that the hypothesis of a major bathymetric effect in the splitting was first proposed by Madelain (1970) and that, if a hypothetical homogeneous MO can be split into veins, this is a fortiori the case for a heterogeneous MO.

The splitting of the MO has been studied intensively in the 1970's and Howe (1982) did a detailed, complete and objective review, saying "it is now more readily accepted that the bottom topography in the Gulf of Cadiz can no longer be regarded as being wholly responsible for the inhomogeneity in the structure of the MO", which clearly disagrees with the much later analysis of Baringer and Price (1997a,b), hence with the assumptions still made nowadays. Also note the result of the chemical-physical pioneering strategy of Howe et al. (1974) who

hypothesized that "the source of the Upper Core might be more appropriately associated with depths of 150-300 m within the Strait that are significantly different to those from which the MO usually descend".

Since these pioneering works about the MO in the Ocean, the work by Ambar et al. (2008) is by far the most instructive and reliable one, thanks to an objective methodology, that has allowed identifying, south of Cape St

Vincent (~36°30'N-8°30'W), two other cores above and below the two major ones. Even though we will only retain the $\sigma_\theta$ range (possibly representative of, or at least related to, the $\sigma_\theta$ range upstream), the specified characteristics are: the Shallow Core ($\sigma_\theta$=27.2-27.4 kg.m$^{-3}$, 400-600 m), the Upper Core ($\sigma_\theta$=27.4-27.6 kg.m$^{-3}$, 600-1000 m), the Lower Core ($\sigma_\theta$=27.6-27.8 kg.m$^{-3}$, 1000-1300 m) and the Deep Core ($\sigma_\theta$=27.8-27.9 kg.m$^{-3}$,



1300-1600 m). Note that the density and depth ranges are contiguous and that the average densities differ by
0.15-0.2 kg.m$^{-3}$ over an overall range of ~0.55 kg.m$^{-3}$.

From a numerical point of view, and to our knowledge at least, no study has specifically addressed the MO
splitting, even though it is a major process that links the Mediterranean Sea and the Atlantic Ocean, and drives,
in particular, the clear signature of the Sea in most of the northern Ocean (e.g. Garcia-Ibanez et al., 2015), and
heterogeneity of the MO has only been considered as a boundary condition for surface circulation studies. Peliz
et al. (2007, 2009) have simulated a vertically stratified MO, only specifying the θ and S of three specific layers
(bottom-250 m-200 m-170 m; associated s values could be 29.07, 28.70 and 28.55 kg.m$^{-3}$) but, even though
reference is made to Baringer and Price (1997a,b), the study does not differentiate the southern/offshore edge of
the MO from the northern/onshore one.


We personally got specific results and proposed original hypotheses about the MO dynamics across the Strait
(that will be presented later on) partly from time series collected with autonomous CTDs moored close to the
bottom within the framework of the HYDROCHANGES program (CIESM group, 2002). We also analyzed CTD
vertical profiles collected in 1985-1986 along cross-Strait transects during the Gibraltar Experiment (GIBEX)
that are easily available (MEDAR group, 2002). We have focused on the MO heterogeneity, on both the
horizontal and the vertical as well as over time, and we have put forward hypotheses based on our own
understanding on the Sea functioning and on a possible major -and not considered yet- role played by the
Coriolis effect on the MO itself. However, as an experimentalist, we are convinced that, in general, we can only
propose hypotheses, and that demonstrations need theoretical simulations, so that we have always asked for an
adequate simulation of the MO dynamics. In any event, we think we are presently able to demonstrate something
with just the data analysis made in Sect.3.

**3 Definitive evidence of the MO heterogeneity**

**3.1 General considerations**

As we did in Millot (2013, 2014b), we first present very simple calculations which should, in no way, be taken
for a proper simulation of mixing processes between the AWs and the MWs, more specifically mixing processes
conditioning the MO structure in the Strait. We consider a relatively thick and homogeneous surface layer (the
AWs) above a homogeneous bottom layer of limited thickness (a hypothetical homogeneous MO at the Strait
entrance that will progress along the Strait), each layer being characterized by specific values of both θ and S.
We simulate the mixing encountered by a portion of the MO in the Strait by running means over larger or smaller
depth intervals and we stop the computation for the total mixing when the bottom is reached. We consider values
characteristic of the MO at the Strait entrance in the mid 1980' (θ=12.9°C, S=38.45) and the mixing with the
base of either the NACW layer (θ=13.5°C, S=37.0) or the SAW layer (θ=14.7°C, S=36.2).




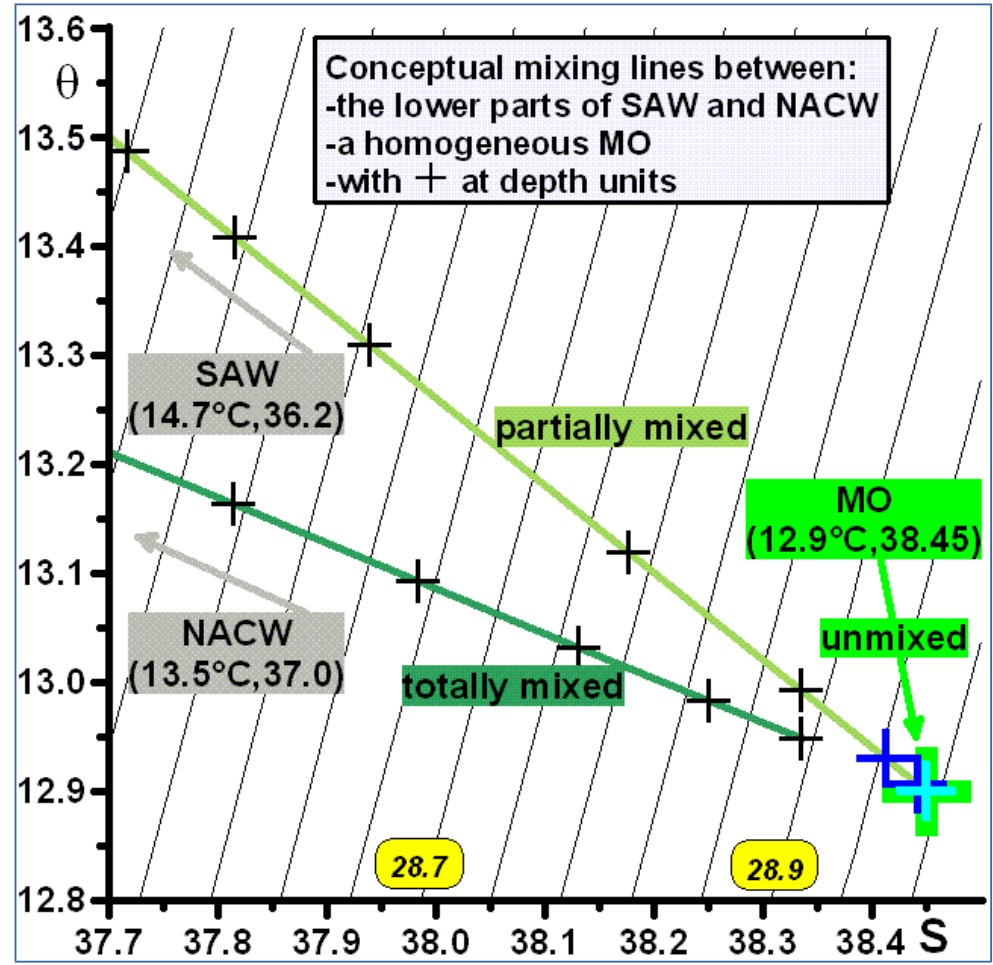

**Figure 1.** Conceptual mixing lines between a homogeneous MO (12.9 °C, 38.45) and the base of either the SAW
(14.7 °C, 36.2) or the NACW (13.5 °C, 37.0) layers schematizing a mixing either partial or total. A single black
cross ends the total mixing line while four (superimposed) cyan and two blue crosses end the partial mixing one.

Considering that there are basically no differences in the two examples of mixing lines in Fig.1, the latter (total
mixing of the MO layer) being just an ultimate stage of the former (partial mixing), and that all results do not
depend in any way of the numerical values, we assume that these conditions (mixing of a homogeneous MO of
limited thickness with homogeneous and possibly different AWs) represent "Stage A". What does "Stage A"
imply?

Assuming "Stage A", then:





a) all mixing lines on a θ-S diagram necessarily converge towards the MO point,

b) a total mixing of the MO layer results in a mixing line ending by a single point different from the MO one,

c) any partial mixing of the MO layer results in a mixing line reaching the MO point, and mixing points accumulating towards the MO point. A direct consequence is that comparing the number of points accumulated "over or nearly over" the MO point (the cyan crosses) with the number of points "nearby" (the blue crosses) allows estimating the thickness and heterogeneity of the still relatively unmixed MO.

Therefore, if "Stage B" (features a to c) is a direct consequence of "Stage A", formulated as "A implies B", and just because of "non-B implies non-A", the non-observation of any of the features a to c over the whole MO implies that the MO is not homogeneous, i.e. is heterogeneous.

Note that a homogeneous MO totally mixed with some AW at a given place, hence leading to a mixing line such

as the one with NACW in Fig.1, can partially re-homogenize downstream, just due to mixing processes induced by interaction with the bottom topography, hence leading to a mixing line such as the one with SAW in Fig.1, the new MO obviously having a different set of q-S-$s_q$ characteristics. In case a unique AW is involved, the new set of MO characteristics will be on the initial mixing line and the MO will still be considered as homogeneous. In case both the SAW and the NACW are involved in the initial mixing, re-homogenization at both places will lead

to two different sets of MO characteristics, hence to a heterogeneous MO. Such a process is illustrated in Part 2.

### 3.2 The in situ data sets

As indicated in e.g. Millot (2008, 2009, 2014a), mixing between the MWs and the AWs during the 1985-1986

GIBEX lead to straight mixing lines essentially between 5°50'W and 6°05'W, both longitudes characterizing the eastern and western sides of the Strait. All mixing lines observed at these longitudes, hence within the Strait itself, during five GIBEX campaigns, will be analyzed in Part 2 but, because we analyze hereafter (Sect. 4 and 5) data collected east from the Camarinal sills (the eastern end of the Strait at 5°45'W) during a GIBEX-Lynch campaign on 4 and 7-8 Nov. 1985, we just consider herein the θ-S diagrams collected during this campaign on

15 Nov. 1985 along a transect at 6°05'W (Fig.2), which is a convenient longitude to check for the possible homogenization of the MO postulated within the Strait. The general characteristics of these diagrams have already been analyzed, over the whole AWs-MWs range and together with diagrams along other transects, in both our 2008 and 2009 papers, been named Lynch3,4; an overview is also given by Fig.1b in Part 2. It is thus obvious that the relatively large slope of the mixing lines (as compared to the ones schematized in Fig.1 with

similar ranges and the same Dq/DS scale) is due to a not well-marked NACW core (θ at the base of the layer of ~14°C instead of ~13.5°C) and a θ at the base of the SAW layer >15°C instead of ~14.7°C). The specific characteristics of these diagrams analyzed at the 1-m/db data level (while immersion of original in situ temperature and salinity data is in db, we computed q and $s_q$ without inferring an immersion in m, hence identifying db and m values) are synthesized below.




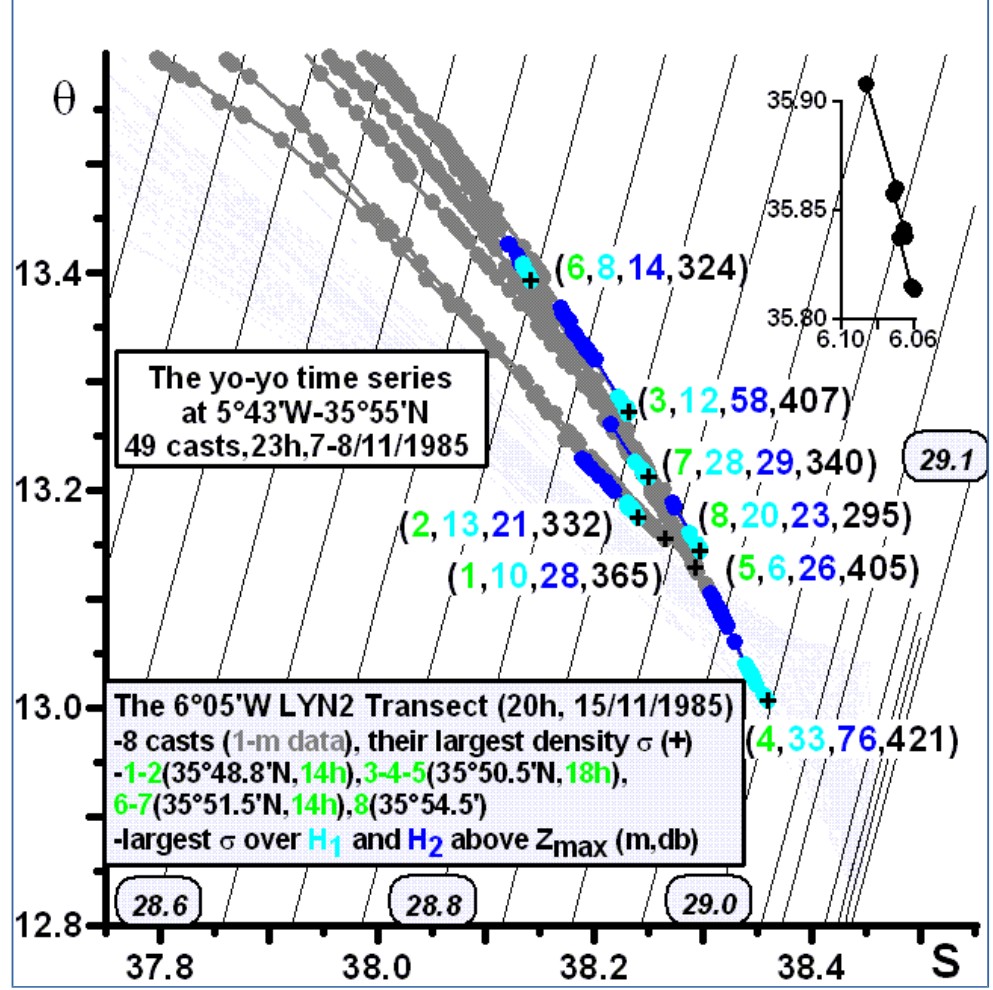

**Figure 2.** Mixing lines at ~6°05'W observed at a 1-m/db level within ~20 hours on 15 Nov. 1985 during the
GIBEX-Lynch campaign. Station numbers are specified in green (#1-2 at ~35°48.8'N, #3-4-5 at ~35°50.5'N, #6-
7 at ~35°51.5'N and #8 at 35°54.5'N) together with the lag between profiles at similar locations, the maximum
depth reached by the profile in black, the $s_{max}$ indicated by the black crosses, and the thicknesses of specific $\sigma_q$

ranges in either cyan or blue. Data have not been colored at #1 and #5 for clarity of the figure but heterogeneity
is similar to that at #2. The yo-yo time series at 5°43'W is plotted in light gray.

First, it is clear that nearly homogeneous waters (the cyan dots) were observed at the bottom on all profiles,
sometimes over relatively large thicknesses (33 m, 28 m, 20 m, etc.), near very different sets of θ-S-σ₀ values

that were distributed over relatively large ranges of ~0.4°C, ~0.2 S units and ~0.25 kg.m⁻³. The facts that the



sampling was relatively coarse and that the transect did not cover the northernmost part of the Strait obviously implies that actual ranges at 6°05'W were larger.

Second, since i) the mixing lines do not converge (clearer examples of non-converging mixing lines are shown in Parts 2 and 3), ii) total mixing is never observed (examples of total mixing are shown in Sect. 4) and iii) partial mixing does not evidence a single point that could be associated with a homogeneous MO (partial mixing evidencing different MWs were generally observed from the Strait entrance (Sect. 4) to the whole Strait (Part 2) and up to the Strait exit (Part 3)), hence since none of the features a-c previously identified are observed, it is definitely demonstrated that the MO cannot be considered as homogeneous, at least in the western side of the

Strait and even more upstream.

More precisely, not only the MO must be considered as heterogeneous, but it must also be considered as a juxtaposition side by side, in a cross-strait direction, of relatively homogeneous and different components that must necessarily result from the set of MWs identified within the Sea interior and at the Strait entrance having

mixed with a set of AWs. Just because the data sets characterizing these homogeneous components are markedly different from the data sets associated with the MWs at the Strait entrance (Sect. 4), one must invoke a re-homogenization process (end of Sect. 3.1) after the intense mixing that occurred at Camarinal sills. All this will be specified in Part 2 and it will be shown that the MO structure in mid-Nov. 1985 was relatively complex; indeed, the MO structure at 6°05'W was unusual and relatively complex on 15 Nov. but it was usual and

relatively simple on Nov. 1-2 (Millot, 2008). Additionally, let us just make a few general comments.

Even though similar locations were occupied a few h apart (profiles # 1-2, #3-4-5 and #6-7; profiles on cross-Strait transects are, in all our papers, numbered from south to north), only #1-2 have sampled nearly the same component of the MO; also note the similar slope of mixing lines #1 and #2, as well as the fact that the slopes in

the south are lower than the others, which is a quasi permanent and general feature all along the Strait. The MO components sampled by either #3-4-5 or #6-7 display a marked variability while the northernmost #8 does not show specific and/or extreme characteristics, which are not general features. In the same way, the fact that the largest density is observed at the deepest (#4) location, as well as the fact that the maximum density at a relatively central location (#3, 407 m) is lower than that at locations more to the north (#7, 340 m and #8, 295 m)

are other non-general features.

Whatever the difficulties in understanding such a set of θ-S diagrams without any additional information, they can in no way be obtained from a MO that would have been homogeneous upstream, except if one imagine a complex series of mixing+re-homogenization processes with different AWs. We lack convenient data sets and

numerical simulations able to validate or reject the hypotheses we have proposed about the mechanisms that, most probably, drive such a heterogeneity of the MO, more precisely about the processes that lead a set of MWs overlying in the Alboran to organize, all along the Strait, into a MO that would have features coherent with those evidenced by such a set of θ-S diagrams.



However, even though we already made a large use of the GIBEX data sets collected along repeated cross-Strait
transects essentially, we recently realized that extremely valuable data sets were also obtained during yo-yo CTD
time series (collected at specific locations) that have never been previously analyzed. Considering our own
present understanding of the MO heterogeneity, we do think that such time series, even if performed during just
one-day periods, are much more valuable than any cross-Strait transect that would be repeated for

days/weeks/months. The major basic reason is that, in an area where the spatio-temporal (x, y, z, t) variability is
so large, any efficient sampling of a parameter P must reduce the number of variables (in this case fix both x and
y) and thus consider $P(x_0, y_0, z, t)$. It is this certitude that i) has always led us to essentially work with arrays of
current meters moorings set in place for up to a year (we have provided 24% of the number of current time
series, 31% of the number of current data and 58% of the total duration of current data -that represents ~105

years- available in early 2017 in the French SISMER data base for the Mediterranean Sea), such arrays being
just completed by campaigns, and ii) has led us to initiate the HYDROCHANGES program (CIESM group,
2002). We demonstrate the value of such yo-yo CTD time series in all three parts of our trilogy.

**4 General results from the GIBEX yo-yo CTD time series at 35°55'N-5°43'W and CTD transect at 5°40'W**

During the GIBEX-Lynch Cruise 702-86 conducted in Nov. (1-17) 1985 between 5°10'W (the western Alboran)
and 6°05'W (the western side of the Strait), CTD profiles were mainly collected along north-south transects
(such as the one analyzed in Fig.2; we previously analyzed all these transects), but others were also collected as

a yo-yo time series at 35°55'N-5°43'W (Fig.3) while having never been analyzed up to now; another such not-
analyzed-yet yo-yo time series was obtained 7 months later at 35°50'N-6°05'W and is analyzed in Part 2. The
former consisted in 49 profiles (nominal/most frequent maximum depths of 250-300 m) and was performed in
~23 h on 7-8 Nov. which, assuming the sampling of a given portion of the MO at various longitudes is more
efficiently done downstream, explains the choice we made of the transects shown herein; indeed, this time series

was performed after a cross-Alboran transect at 5°40'W (on Nov. 4, Sect. 4.2) and before the cross-strait transect
at 6°05'W (on Nov. 15, Sect. 3). Note that this time series is located east of the Camarinal sills (5°45'W), within
the zone we define as the Strait entrance, and on the upper southern continental slope where conditions are
expected to markedly differ from those over the northern slope (see Sect. 6).

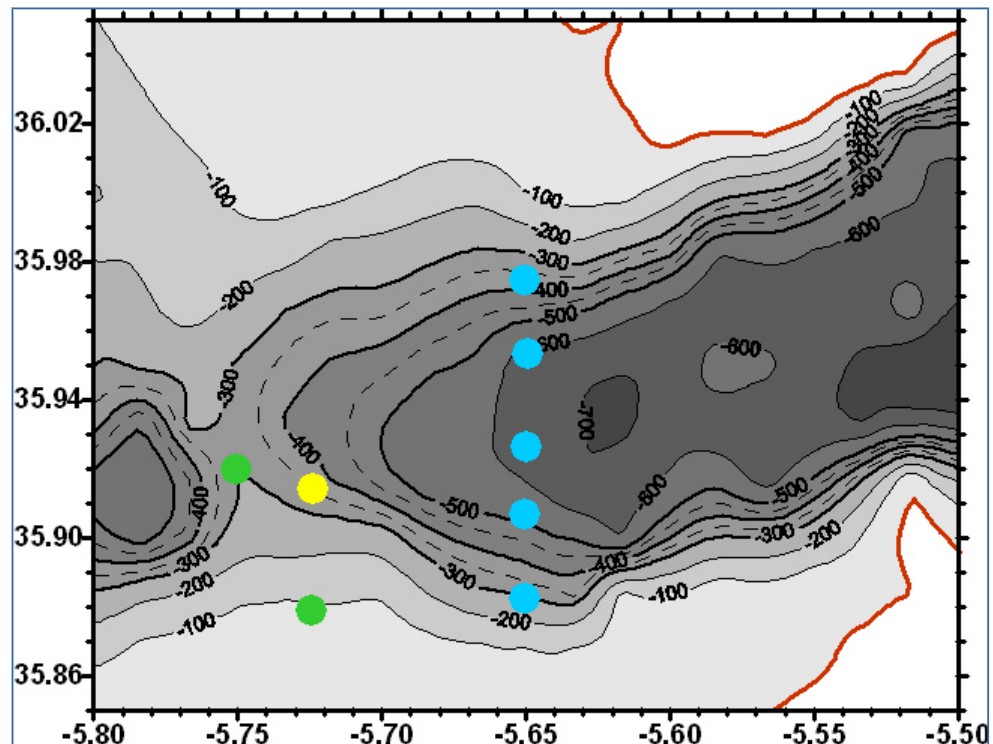

**Figure 3.** Relatively rough bathymetry (the northern and southern Camarinal sills are not identified) of the studied area with the five main locations of the profiles along the 5°40'W transect (blue) and the nominal location of the 49 profiles during the yo-yo time series at 35°55'N-5°43'W (yellow), both from the GIBEX-Lynch campaign in Nov. 1985, and the locations of the two HYDROCHANGES CTDs moored at 270 m and 80

m, in particular from 2003 to 2007.

Our overall aim is to make an analysis as objective as possible. We first (Sect. 4.1) argue for the classification of these 49 profiles into five groups on the sole basis of maximum density (of each profile) ranges and we specify some characteristics of these groups that we will identify with specific colors, namely blue, violet, pink, red or

orange with decreasing density. We previously associated these colors with given MWs (e.g. Millot 2009, 2014a), but have the feeling that this was not considered as fully objective and prevented from a rapid agreement with our ideas, so that we avoid dealing with names and deal only with colors in Sect. 4 and 5, reference being possibly made to Table 1.


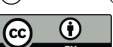


**4.1 The maximum density ranges associated with the yo-yo time series and their associated characteristics**

The θ-S diagram in Fig.4a represents the largest $\sigma_{max}$ values for 43 out of 49 profiles. Three groups, separated by
nominal isopycnals of 29.095 and 29.090 kg.m$^{-3}$, can be clearly identified and the blue group is more widely
distributed than the pink and violet ones. As shown later on, the violet cross that is relatively isolated from the
rest of the group and is the closest to the 29.095 kg.m$^{-3}$ isopycnal corresponds to profile #20 that was sampled
after #6 to #19 in violet and before #21 to #23 in blue: profile #20 thus represents some transition phase between

the violet and the blue conditions, it could have been colored in blue (with a lower separation isopycnal defined
with 4 digits) but it also represents a unique situation since all other crosses clearly belong to one or the other
groups. Note that significant differences in either θ or S cannot be evidenced.

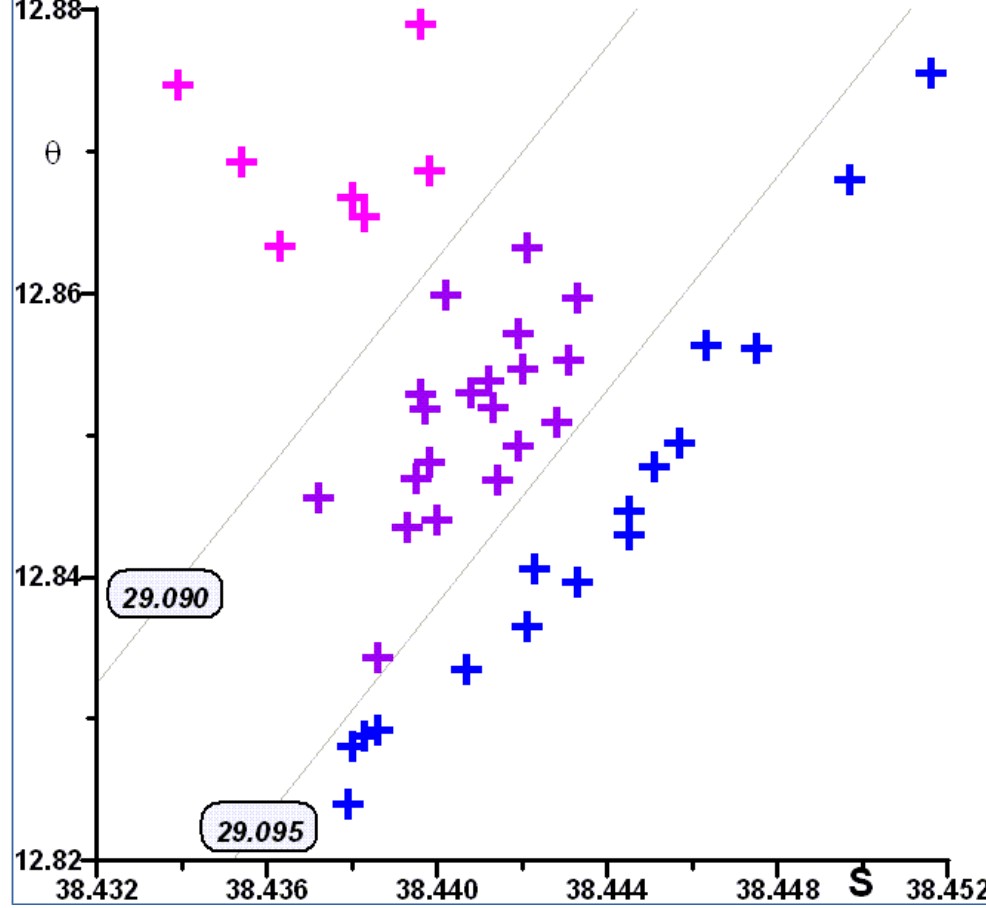

**Figure 4a.** The largest $\sigma_{max}$ associated with 43 (out of 49) profiles from the yo-yo time series. Coloring is
proposed according to round values of convenient separating isopycnals.





The significance of these three groups can be illustrated by the histograms in Fig.4b. The red histogram represents the 49 $\sigma_{max}$ values (47 in the displayed $\sigma_q$ range) and minima are clearly observed for the 29.094-

29.096 and 29.089-29.091 ranges, which accounts for the two separation isopycnals we defined. The blue histogram represents all values from the 49 profiles (5588 in the displayed $\sigma_q$ range) and a minimum is still clearly observed in the 29.093-29.094 range. Even though there is a natural shifting of the two red major maxima (and of the minimum in-between) towards similar blue features at lower $\sigma_q$ values, the two blue pics are more significant than the red ones and they clearly account for the occurrence of at least two very different dense

MWs. The histogram in green represents ~28000 hourly values (out of ~36000, i.e. ~4 years) from the HYDROCHANGES time series at the Camarinal southern sill (5°45'W, 270 m) in 2003-2007 and already analyzed (in e.g. Millot (2009) and Millot and Garcia-Lafuente (2011)). Not considering the long-term changes between the 1980's and the 2000's, larger $\sigma_q$ values are obviously observed during such a relatively long period. The absence of separated maxima is partly due to the shifting towards lower values since MWs at the sill are

more mixed, both together and with the AWs, than at 5°43'W (just 0°02' upstream); but very different MWs encountered during periods of months and even years (Fig.22 of Millot, 2009) contribute to the smoothing. No doubt that CTDs moored or yo-yo time series repeated there, as well as at the same longitude and depth over the northern slope (see below) would be extremely valuable.

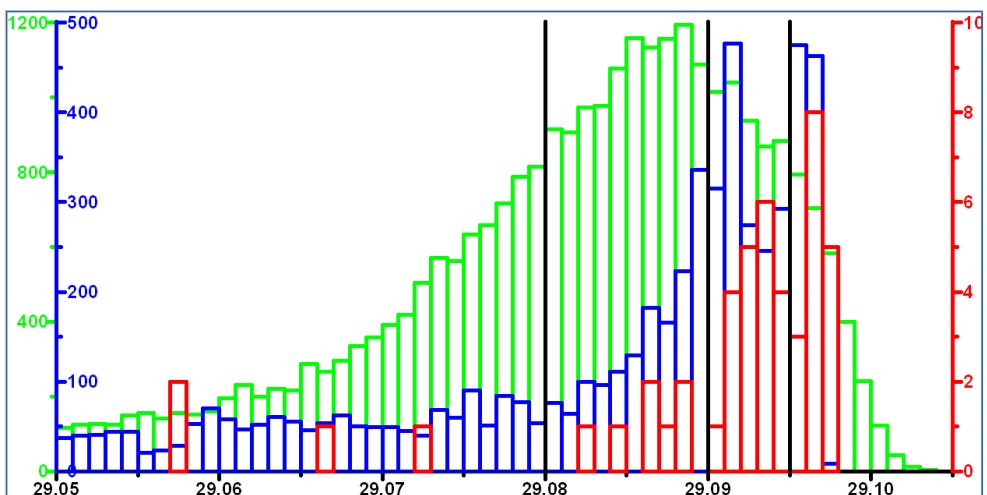

**Figure 4b.** Histograms of the 49 $\sigma_{max}$ values (red) and of the overall largest $\sigma_q$ values (blue) from the 15 Nov. 1985 yo-yo time series at 5°43'W together with the histogram from the 2004-2007 hourly time series at the Camarinal southern sill (5°45'W, green).





In order to display the whole range of the 49 maximum densities $\sigma_{max}$ in an as clear as possible way, and noting that the overall largest density is 29.0975 kg.m$^{-3}$, we define for each profile a $D\sigma = (29.0976 - \sigma_{max})$ kg.m$^{-3}$ and we plot (Fig.5) the function $-\log D\sigma$ with respect to z (Fig.5a), q (Fig.5b) and S (Fig.5c).


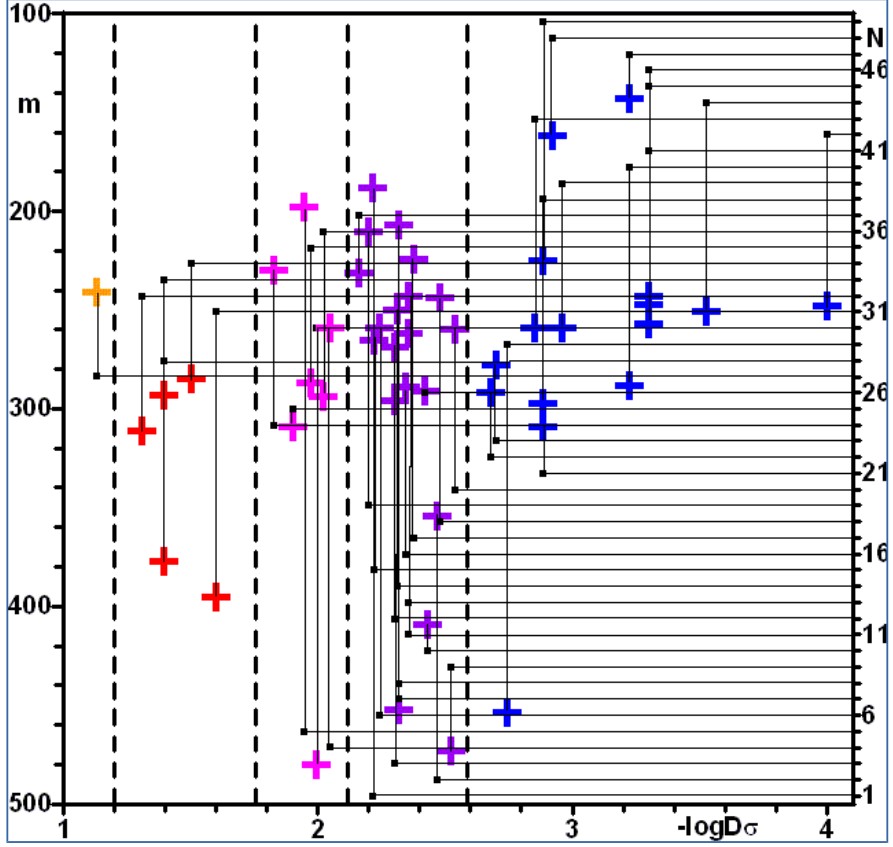

**Figure 5a.** Distribution of $\sigma_{max}(z)$ plotted as a function of $-\log D\sigma$, to allow a convenient representation of the

whole $\sigma_q$ range, with $D\sigma = (29.0976 - \sigma_{max})$ kg.m$^{-3}$, together with the data numbering and the isopycnals coloring we propose for the grouping into five groups.

Considering all three Fig.5a-c, it appears that the two other separation densities we propose (29.080 and 29.035

kg.m$^{-3}$), chosen at "3-digit round values" objectively considering both the crosses distribution as well as our (possibly non-objective) knowledge and understanding of the sea functioning, can be reasonably accepted. Note that the separation density between the red and pink groups could have been chosen in the 29.075-29.800 kg.m$^{-3}$ range but, essentially because the pink group is more compact than the red one, we chose the value closest to the pink group, which allows a wider range for the less well defined red group. Additional arguments will be

provided both hereafter and in Part 2.



Fig.5a represents the depth at which $\sigma_{max}$ (for each of the 49 profiles) is observed as a function of -logD$\sigma$; note that the intermediate groups (violet, pink, red) that have density ranges of 0.005 (29.095-29.090) kg.m$^{-3}$, 0.010 (29.090-29.080) kg.m$^{-3}$ and 0.055 (29.080-29.035) kg.m$^{-3}$ have roughly similar widths with such an x-scale unit.

It is clear that the blue/densest group can be observed not only at the 250-300 m nominal depth and at relatively large depths of ~450 m (which could be considered as normal but which, according to our previous studies and Sect. 5 for instance, is certainly not the most frequent situation) but also at relatively shallow depths of ~150 m (which does not surprise us). Moreover, the shallowest $\sigma_{max}$ are among the largest, which is actually the situation we mostly expect (see Sect. 6) and that, even if there are only few crosses in the 400-500 m depth range, the

deepest is pink.



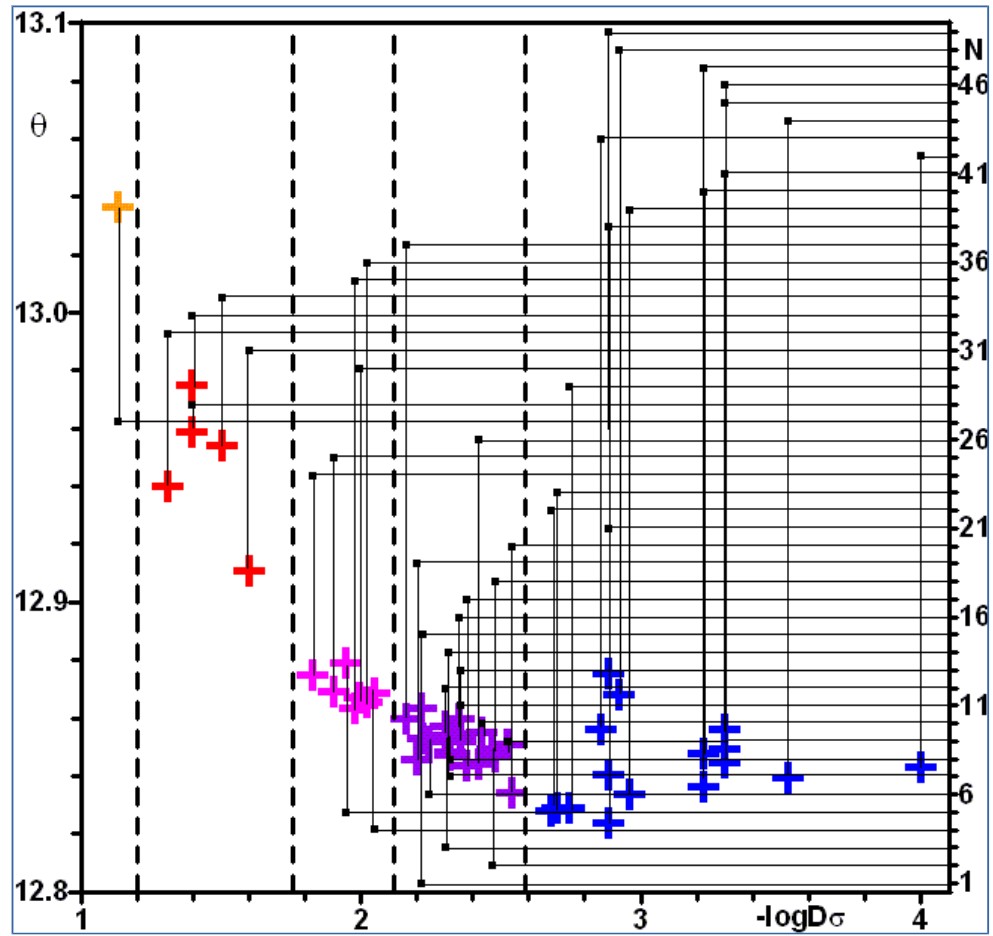

**Figure 5b.** Same as in Fig.5a for $\sigma_{max}(\theta)$.





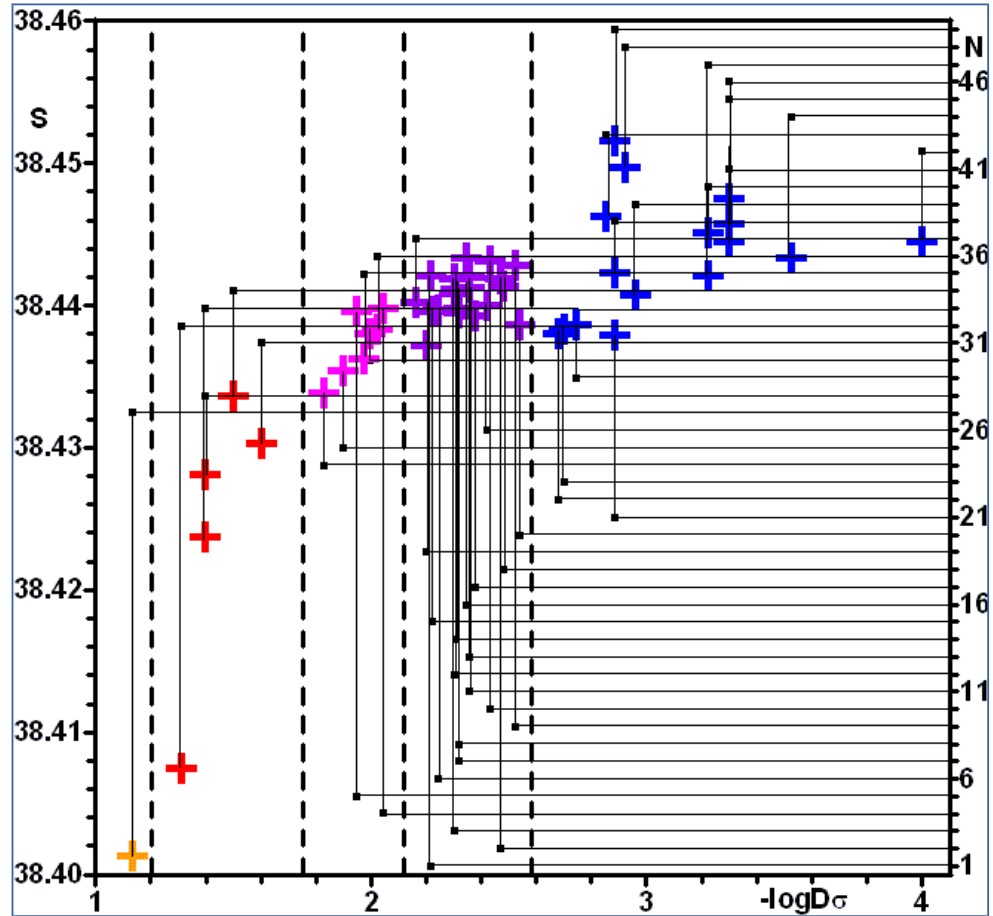

**Figure 5c.** Same as in Fig.5a for $\sigma_{max}(S)$.

Fig.5b and Fig.5c represent the relationship between $\sigma_{max}$ and the associated θ and S values, respectively (in
complement to Fig.4a). The major information provided by Fig.5b,c is that the pink and violet groups are
significantly different since the pink is markedly warmer and fresher than the violet. However, both are relatively
compact, clearly evidencing two different groups of relatively homogeneous MWs that are neither the coolest
nor the saltiest, even when considering the groups' barycentres. Intuitively, one can think to relatively old and
markedly mixed MWs that, therefore, have followed a relatively long -in time and/or distance- route and/or
intense mixing processes since they left their zone of formation.

Comparatively, and even if the blue group corresponds to a much smaller density range of 0.0025 (29.0975-
29.095) kg.m$^{-3}$, associated θ and S spread over ranges much wider than those associated with the pink and violet
groups, while θ and S blue values can be either larger or smaller than the pink and violet ones. Such a spreading



can only be the characteristic of a relatively young and unmixed MW that, therefore, has followed a relatively short -in time and/or distance- route and/or relatively reduced mixing processes since it left its zone of formation. Note that the coldest or saltiest values encountered in the blue group can hardly result from some mixing only involving the other groups, except if one assumes that, from the zone where the blue water was formed, the other waters, that would have necessarily been coolest or saltiest there, have then encountered more intense mixing.

Also note that the coolest values are not the saltiest, that the former are lighter than the latter, and that the densest are associated with medium θ and S values. Intuitively, one can thus think that the relatively young and unmixed blue MW has encountered intense cooling (latent and sensible heat losses) and salting (due to evaporation) at relatively small scale only as a result of air-sea interactions, as is generally expected during the winter in zones of dense water formation.


As for the orange and red groups, they are the warmest and freshest of all groups. Even if they are composed of only one and five profiles, resp., their characteristics and differences are supported by all the other orange and red θ-S sets encountered at intermediate depths on most of the 49 profiles, as well as along the 5°40'W transect. Even if the red group is composed of only five profiles, the spreading in both θ and S is larger than those

associated with the pink, violet and even blue groups that have 7, 20 and 16 profiles. Consequently, as we suggested for the samplings at 6°05'W in Fig.2, it is very probable that the larger the number of additional profiles at the same place, the wider the spreading of both θ and S in the orange, red and blue groups, as compared to the pink and violet ones.

We will show that $\sigma_{max}$ is sometimes observed far above the maximum depth reached by the profile, which is one feature (among others illustrated hereafter) accounting for the occurrence of intense mixing processes, even at relatively large depths. Let us also specify that the bottom depth indicated in the data files (always in the 250-300 m nominal range) is, as usually, the one measured at the beginning of the (downcast) profile, but that drifting of the ship (specified from radar measurements -no GPS at these times) across the continental slope during the

profile sometimes allowed sampling depths as large as ~500 m. Also, at least to our knowledge, the GIBEX CTD probes were not equipped with an acoustic device that would have allowed sampling actually down to the bottom; lowering of the CTD was guided by depth sounder information so that the especially difficult conditions in the Strait did not allow sampling the deepest parts of these homogeneous layers.

Before providing a detailed analysis of these 49 profiles from the yo-yo time series at 5°43'W (Sect. 5), and just because some of them display very strange or at least features we did not expect, we first analyze each of the six profiles from a transect at 5°40'W that display more classical features (Sect. 4.2). However, on each of the profiles from both the transect and the time series, accumulation of data (points on a θ-S diagram looking like the cyan ones in Fig.1 and 2), hence relatively homogeneous layers, are observed not only close to the bottom, as

most of the time, but also over the whole depth range (below ~50 m). It is thus clear that "the whole layer of all MWs" at both places is in fact mainly composed of "up to five relatively homogeneous layers", each relatively homogeneous layer being associated with one specific MW, while a more or less intense mixing between two



such overlying layers leads to a more or less thick "interface layer in between". The maximum density of each
relatively homogeneous layer being generally found near the base of that layer and being often well within one

of the five $\sigma_{max}$ density ranges previously defined, be this layer close to the bottom or at intermediate depths, we
color accordingly each set of accumulated data; note that, most of the time, the minimum density of such
relatively homogeneous layers is out of the specific range associated with the color. All profiles have been
analyzed individually at the 1-m/db data level, and all q-S diagrams and vertical profiles are presented with the
same θ-S-$\sigma_\theta$-z ranges.


**4.2 The 5°40'W transect**

The thickest and most homogeneous layer at #1 (Fig.6a) is red (23 m near 270 m), the deepest layer (10 m) close

to the bottom at only ~320 m is pink and numerous small scale (few m) layers above ~250 m are orange. The
overall mixing line is relatively straight between the densest MW and the AWs, which is a general features of all
southernmost profiles at the Strait entrance. Slight but significant instabilities are evidenced in the red layer.



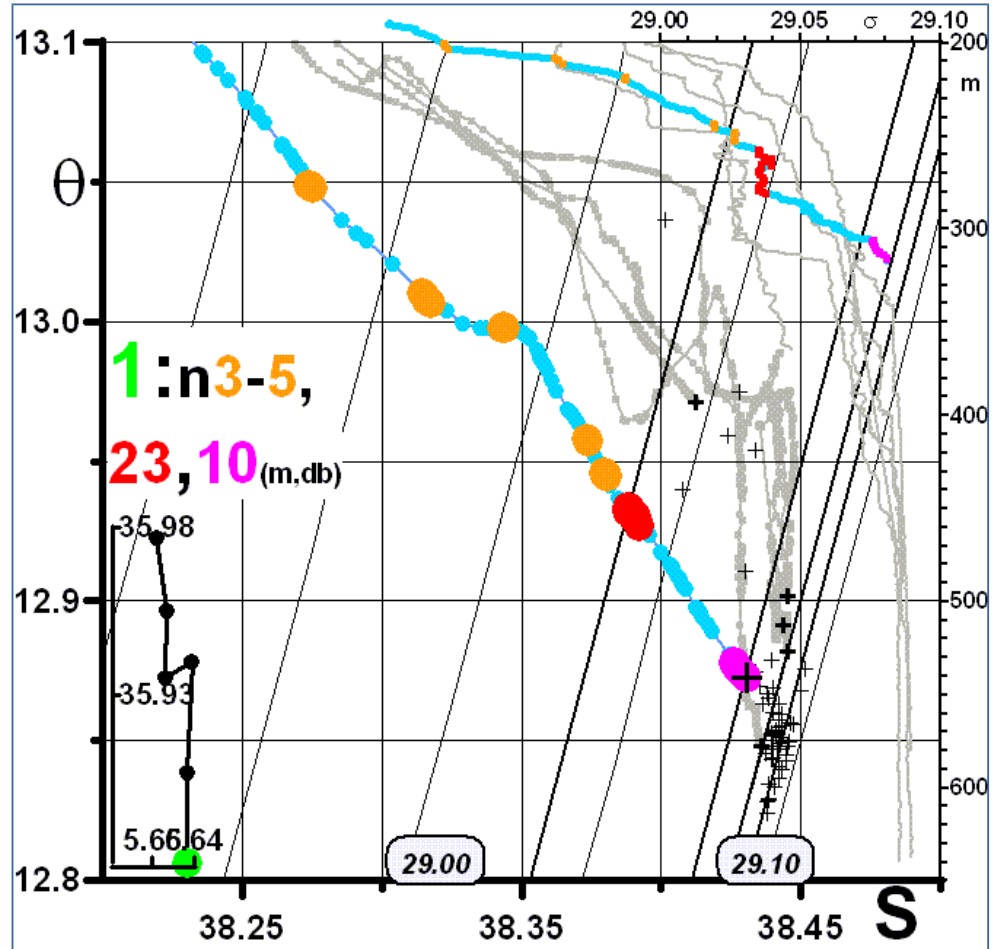

**Figure 6a.** For profile #1 at 5°40'W: θ-S diagram (ranges of 12.8-13.1 °C and 38.2-38.5) of all profiles (gray)
with data plotted at the original 1-m/db level, of the specific profile (light blue), together with inserts displaying
the latitude-longitude position (obtained by radar) of the profile and $\sigma_q(z)$ with corresponding fixed scales. The
proposed coloring is based first on the visual definition of homogeneous layers from $\sigma_q(z)$ and second from the
specific value of the $\sigma_{max}$ at the base of that layer with respect to the separating isopycnals (29.035, 29.080,
29.090 and 29.095 kg.m$^{-3}$ plotted in thick), be this layer lying over the bottom or at intermediate depths. The $\sigma_{max}$
values for each of the profiles are noted with crosses either thick for those of the transect or thin for those from
the yo-yo time series. The whole $\sigma_q(z)$ is plotted in light blue as a function of depth (as a line) while colored data
are plotted as individual values (as dots), which allows clearly evidencing largest gradients (e.g. the red layer in
Fig.9c).



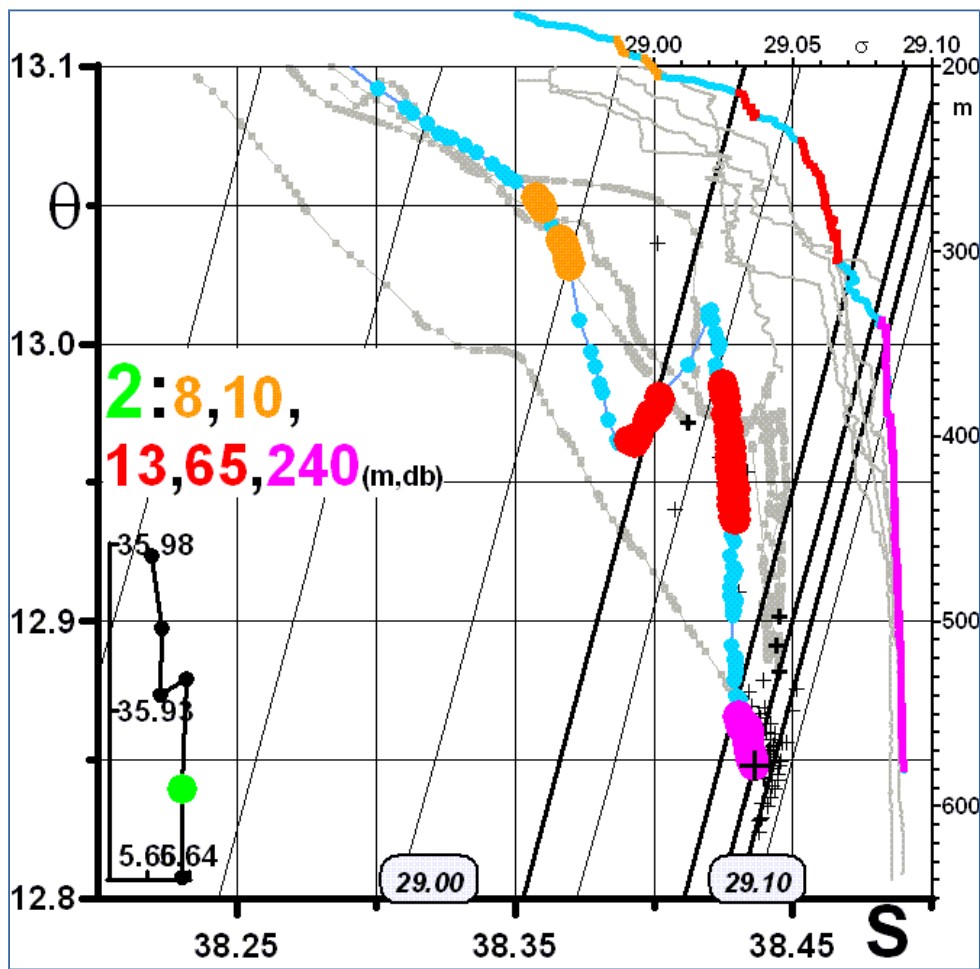

**Figure 6b.** Same as in Fig.6a for #2 at 5°40'W.

The thickest and most homogeneous (over 240 m) layer at #2 (Fig.6b) is the deepest one (down to ~580 m) and,
and it is pink since $\sigma_{max}$ = 29.0898 kg.m$^{-3}$. It is more homogeneous than the two red ones that are 65-m and 13-m
thick and than the 8-m and 10-m thick orange layers above. Note that the overall diagram is "sinuous", depicting

the classical (even with sharper angles) "scorpion-tail image" (Tchernia, 1972) usually found in most of the
western basin. It can also be noticed that the two red layers are on both sides of the tail (consistently with the
remarks made in Millot 2013 and 2014b), hence associated with markedly different slopes of the θ-S diagram.
Mixing of the AWs occurs, as more or less everywhere in both the western and eastern basins (except at the
Strait and the Channel entrances), only with the lightest (orange) of the MWs. Note the marked differences in the

$\sigma_q(z)$ slopes between the homogeneous layers and the interface layers in between.





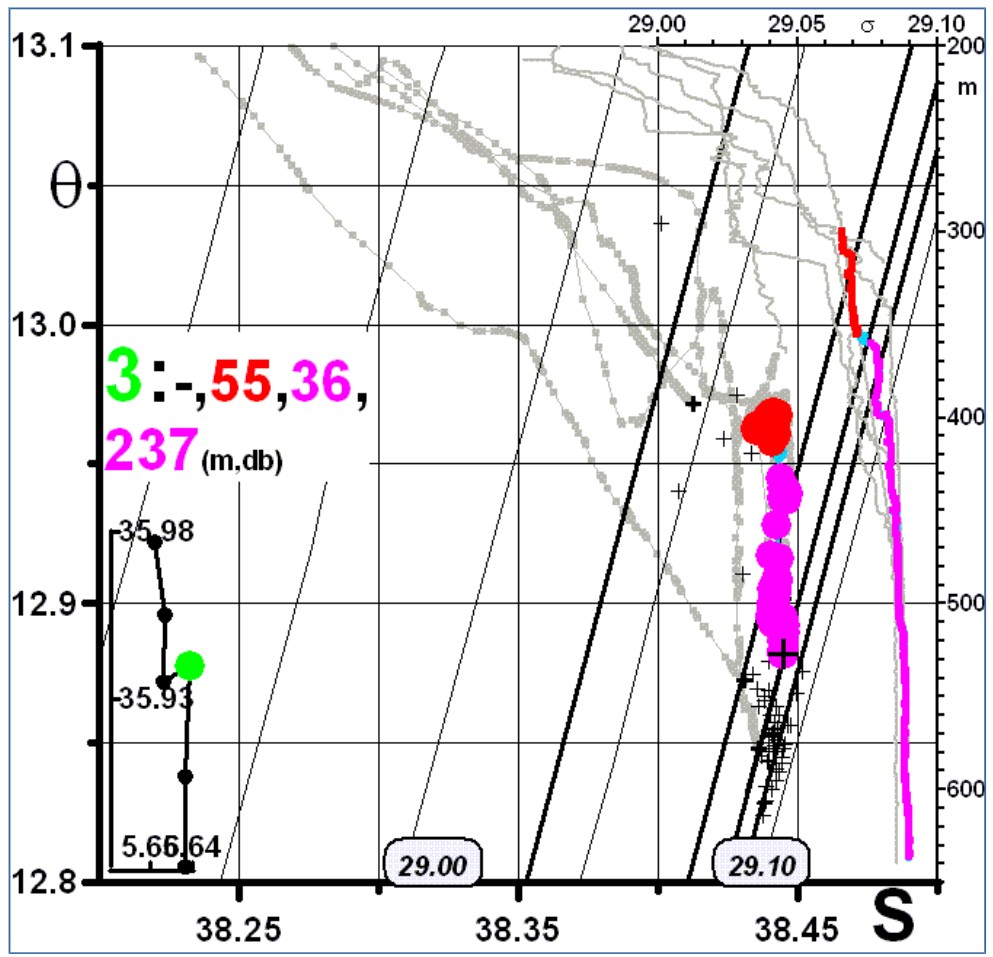

**Figure 6c.** Same as in Fig.6a for #3 at 5°40'W.

Since #3 (Fig.6c) was performed below ~300 m only, the orange and eventually upper-red layers were not

sampled while the deeper-red layer might not have been entirely sampled. A deep layer (maximum depth ~640

m, thickness ~237 m) is colored in pink since $\sigma_{max}$ = 29.0899 kg.m$^{-3}$, but another pink layer of 36 m must also be

considered.

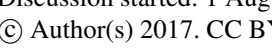


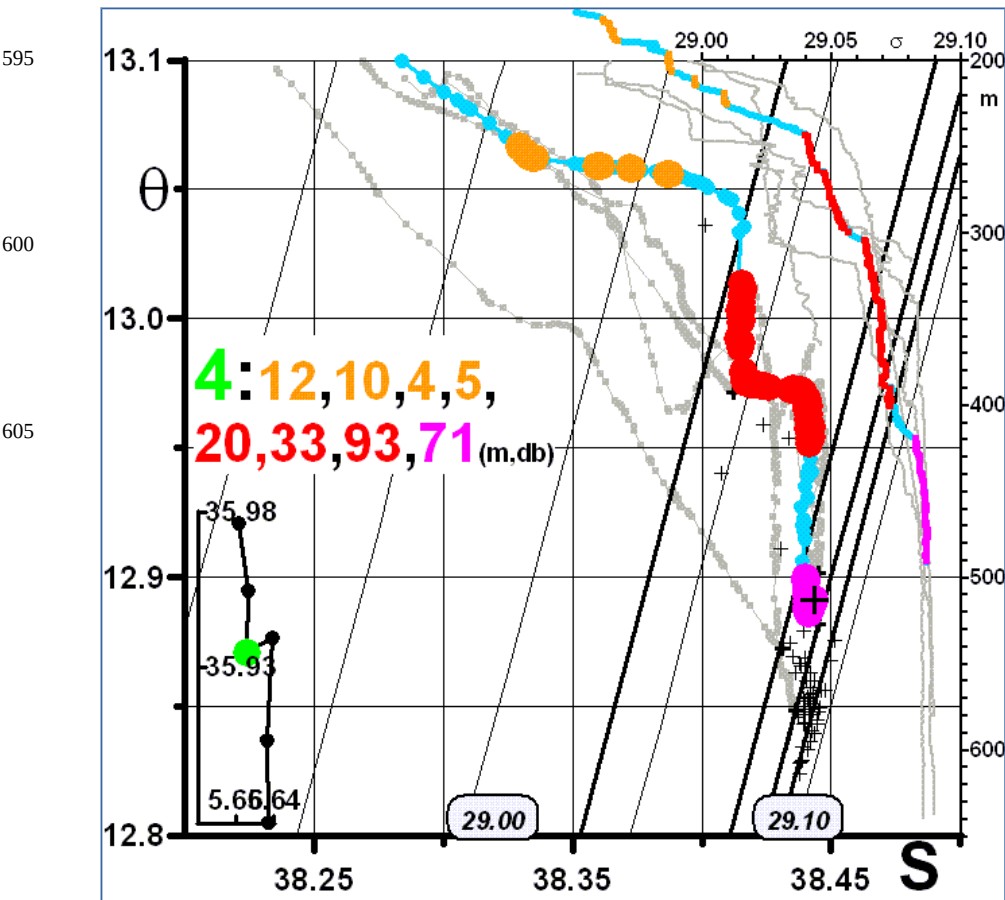

**Figure 6d.** Same as in Fig.6a for #4 at 5°40'W.

Profile #4 (Fig.6d) was performed just to the south of #3 (Fig.6c) and it reached a maximum depth of only ~490 m (~640 m at #3), evidencing pink water below ~400 m. Such differences account for a significant spatial and/or temporal variability and illustrate the necessity to reduce, as much as possible, the sampling interval. A relatively thick set of three red sub-layers (20, 33 and 93 m), that give to the diagram an unusual angled shape, and several few-m orange sub-layers are observed at lower depths.





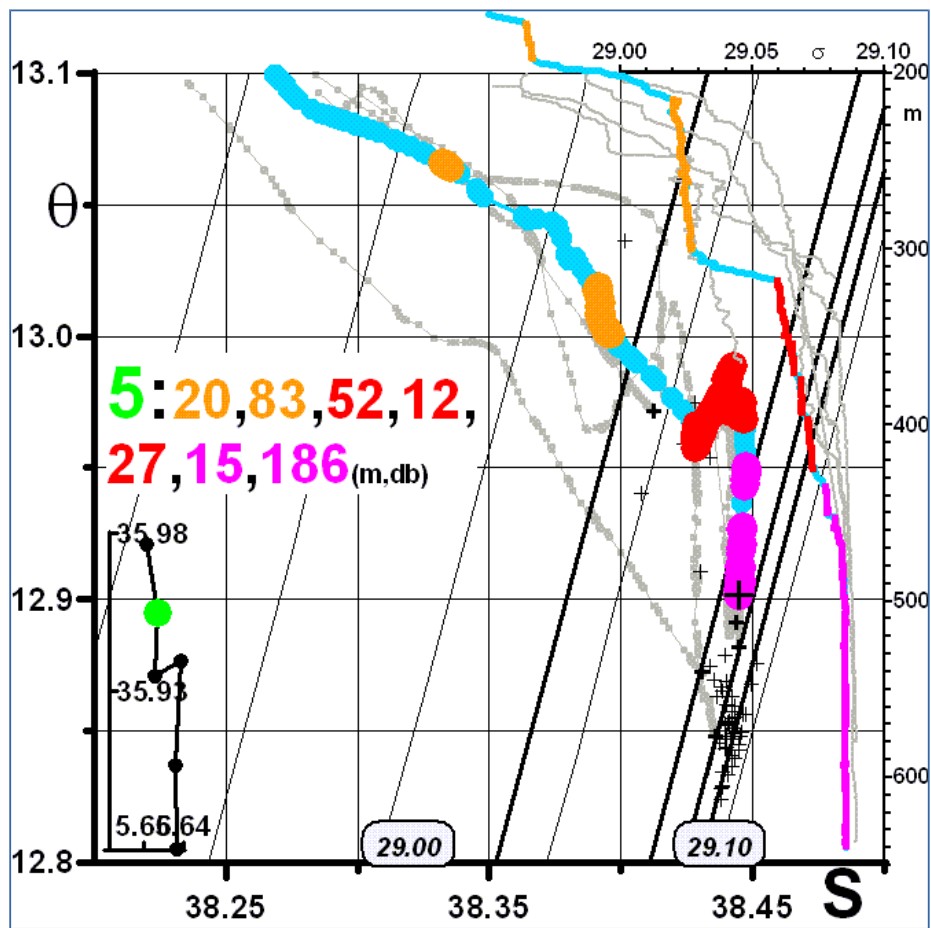

**Figure 6e.** Same as in Fig.6a for #5 at 5°40'W.

Profile #5 (Fig.6e), as deep (~640 m) as #3 a bit more to the south, clearly evidences a ~200-m pink layer. As for

#4, three red sub-layers are observed over ~90 m and associated segments on the θ-S diagram have markedly

different slopes. A major difference with previous profiles more to the south concerns the orange density range

that depicts a very thick (83 m) layer at 200-300 m and another still relatively thick one (~20 m) near 180 m.






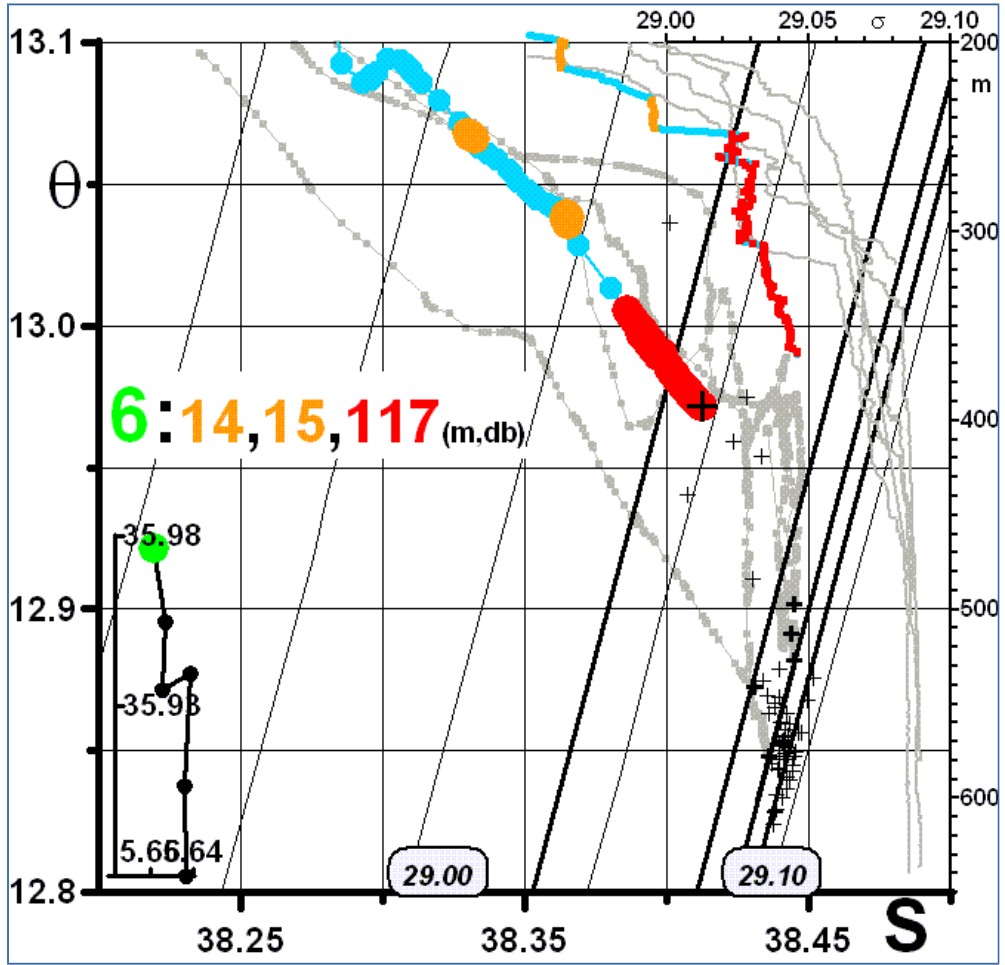

**Figure 6f.** Same as in Fig.6a for #6 at 5°40'W.

On the upper northern slope (#6, Fig.6f), a thick (117 m) red layer, that could be decomposed into three sub-layers, is found over the bottom (~360 m); note that even though the q-S diagram is relatively straight, $\sigma_q(z)$

clearly evidences instabilities over more or less large vertical ranges within each of the sub-layers. Two relatively thin (14 and 15 m) but clearly defined orange sub-layers are evidenced above.

The coloring above-defined is reported in Fig.7. For simplicity, we did not segmented the various color ranges as evidenced from the θ-S diagrams and the $\sigma_q(z)$ curves. This is partly because we did not find strong coherency

between these details from one profile to the other but also, and mainly, because we are personally convinced that the non-simultaneity of nearby profiles performed ~1 h and ~2 km apart prevent from finding coherency at such small vertical scales. However and overall, we are convinced about the significance of the following



features: the red and orange MWs are concentrated more to the north, so that the pink waters are concentrated more to the south, and there is an overall north-south sloping of the interfaces between the various MWs. As for

the densest MWs, they are essentially cooler (by ~0.02 °C) and fresher (by ~0.02 S units) in the south; but they are also denser, as indicated by the isopycnal north-south slope over all these MWs that is consistent with the isothermal and isohaline slopes at intermediate depths. Whatever the case, no blue MW was observed with such a sampling on 4 Nov. at 5°40'W. Assuming the blue and densest MW could not have flown in between the various profiles, hence having been missed by the transect, it must be concluded that either it was not present

anywhere across a shore-to-shore (Morocco-Spain) transect or that it was either northward from #6 or southward from #1. We let the reader open to any hypothesis for the time being.





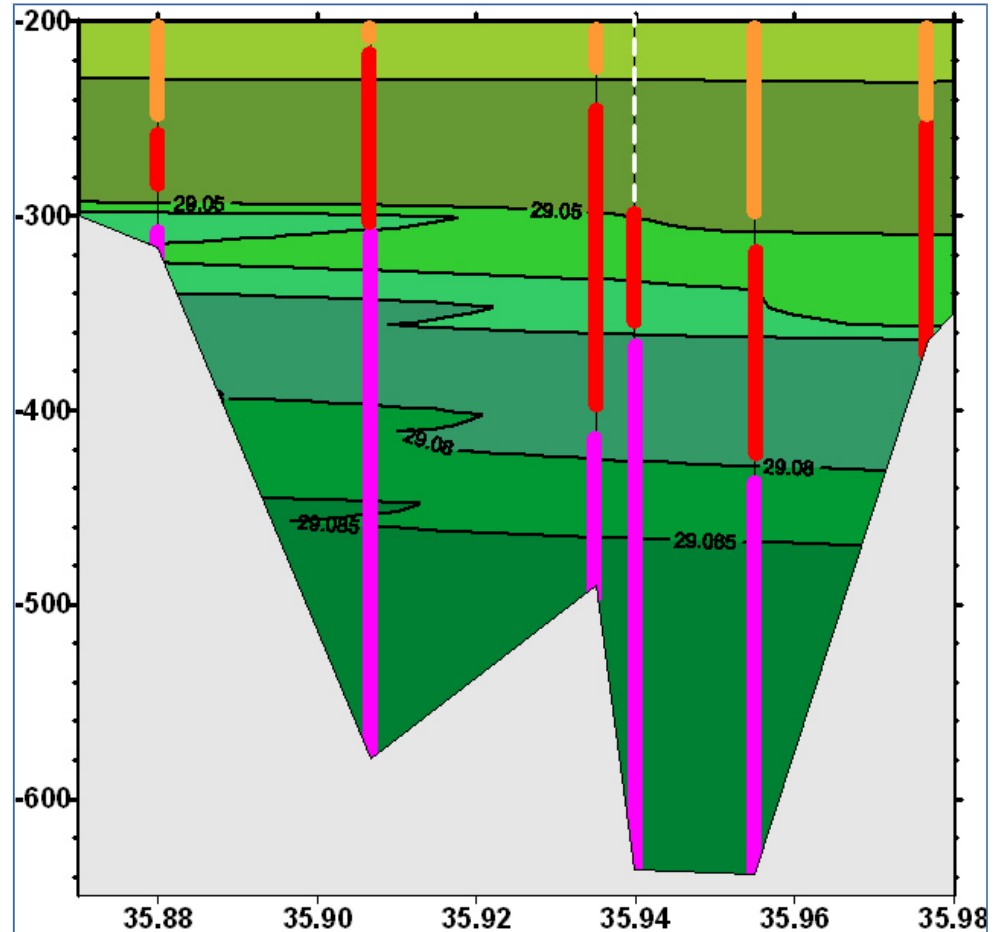

**Figure 7a.** Density $\sigma_q(z)$ distribution across the 5°40'W transect together with the rough overall coloring specified from Fig.6 (sub-layers of a given color are not specified).





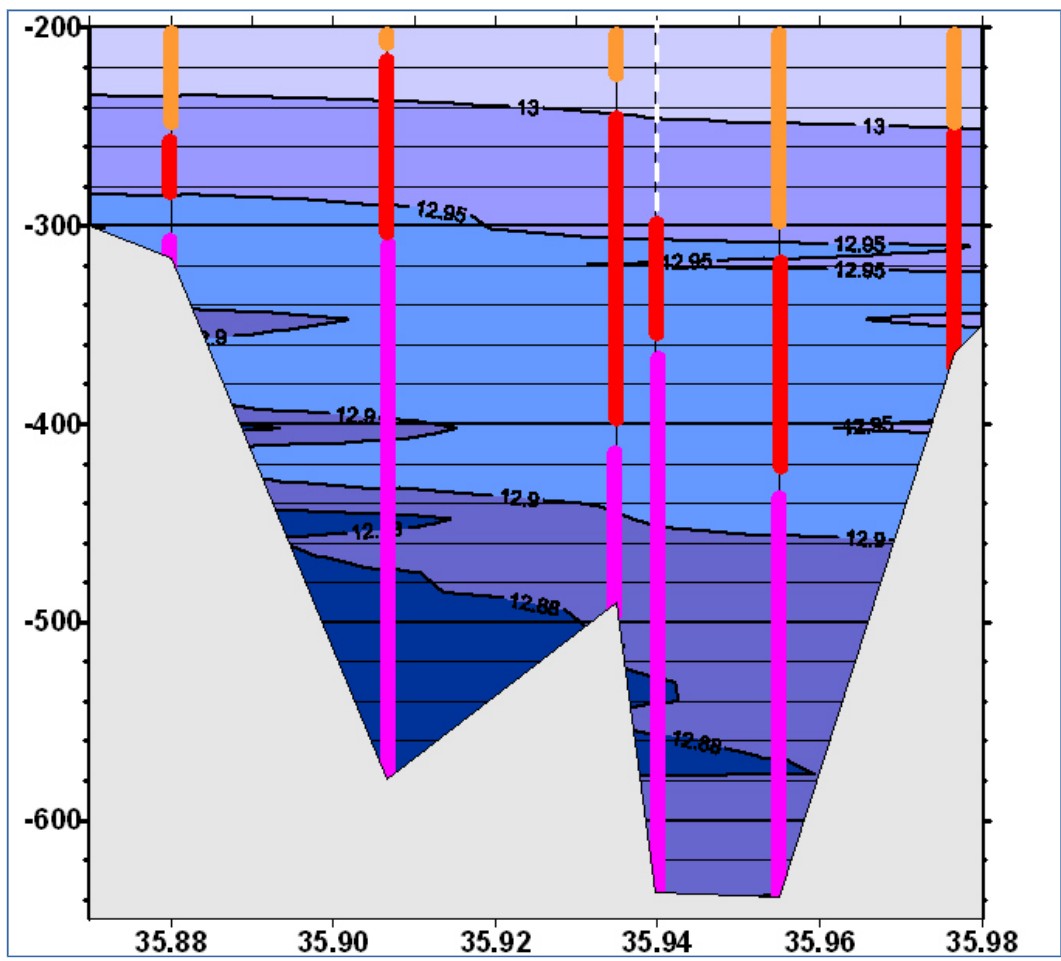

**Figure 7b.** Same as in Fig.7a for θ.



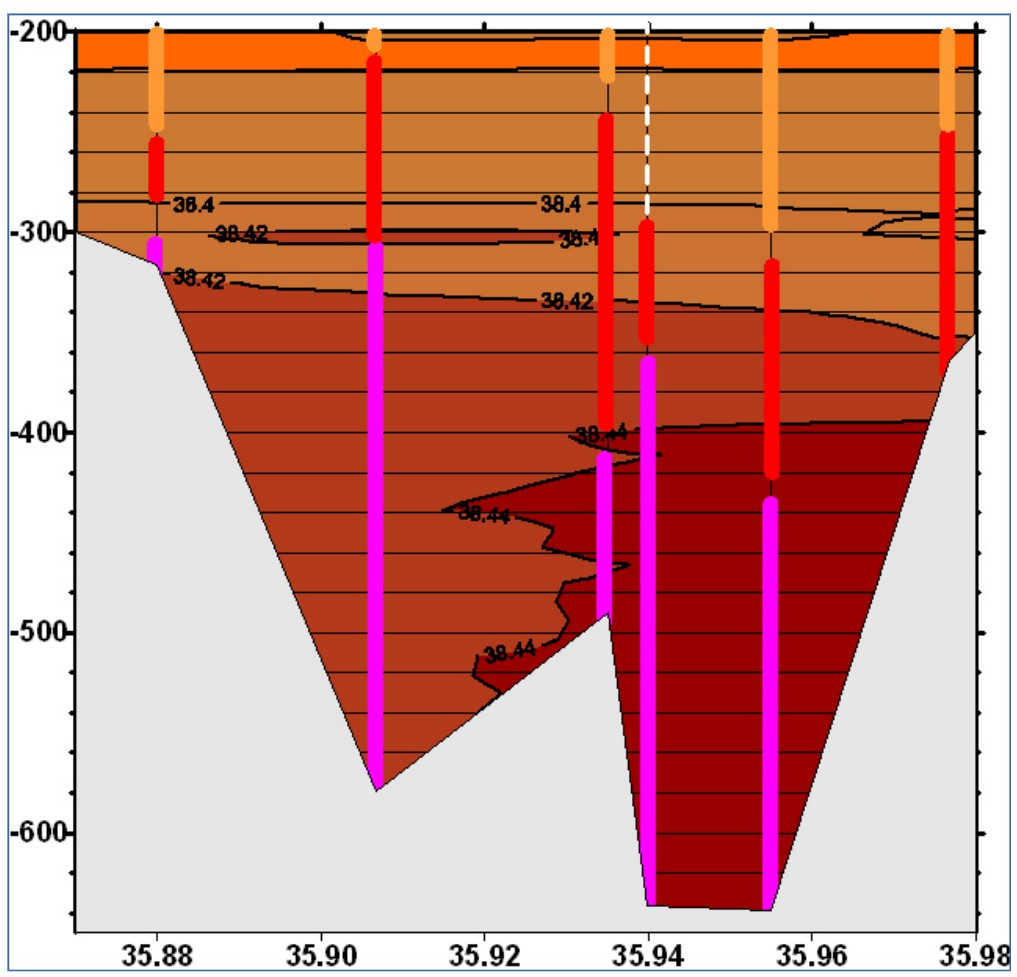

**Figure 7c.** Same as in Fig.7a for S.

 

**5 Detailed analysis of the GIBEX yo-yo CTD time series at 35°55'N-5°43'W**


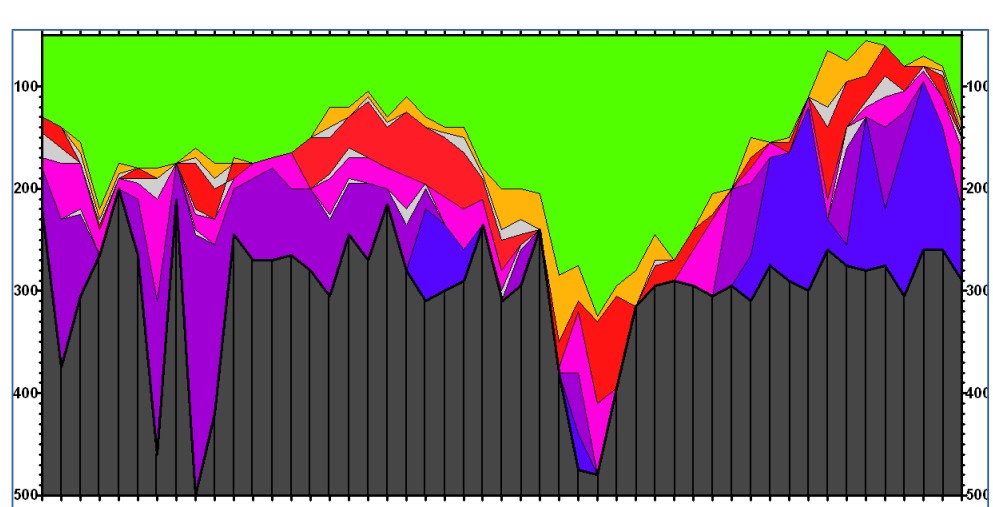

**Figure 8.** Distribution over time (as profiles #) and depth (m/db) of the five colored waters together with interface layers between the MWs in gray and AWs in green.

In order to better appreciate the specificity of each of the θ-S diagrams shown either hereafter or as Supplementary Information (Fig.S1 to S49), we first show the synthesis of such a detailed analysis (Fig.8). The 250-300 m nominal depth being reached most of the time (near the actual bottom depth) indicate that either the ship roughly remained close to the nominal position or that it drifted alongslope during the profiles. When it drifted toward greater depths, either only one relatively dense water was mainly sampled below the nominal

bottom depth (as the violet water at e.g. #7, 9), which is the most expected situation, or several if not all waters were sampled there (e.g. #29-30), which is certainly a temporary but realistic feature that just reveals the tremendous variability in the stratification near 5°43'W. Additionally, a semi-diurnal periodicity that such a ~23-h time series cannot properly resolve might occur there or, more probably considering the vicinity of the Camarinal sills, occurs there: this periodicity appears from the maximum and minimum thicknesses of the MWs

overall layer above the nominal bottom depth. The six orange or red profiles numbered between #27 and #34 (Fig.5) are easily identified during the period when the largest offshore / northward drift of the ship and the largest deepening of the MWs was observed: the fact that the lightest MWs were observed at such relatively deep locations illustrates the dramatic effect of the tide and the complexity of analyzing isolated samplings there. It can also be noticed that, roughly, color at the bottom is mainly violet at the beginning of the time series and

blue at the end, while these colors were not clearly (some $\sigma_{max}$ were just below the pink-violet separation) or clearly not (blue) observed at 5°40'W (Fig.6). Even though this synthesis suggests that all 49 profiles are interesting, we can analyze in details only some (6) of them; the others (Fig.S1 to S49) demonstrate our objectivity and the continuity of the specific characteristics from one profile to the other.





Profile #9 (Fig.9a) is interesting because it is relatively deep with nearly all colors displayed; indeed, $\sigma_{max}$ =
29.0946 kg.m$^{-3}$ so that the deepest homogeneous layer (440-505 m) could easily have been colored in blue.
Instabilities revealing mixing processes are evidenced essentially in the violet and red layers while interface
layers are clearly observed between the orange, red, pink and violet ones. Such an overall step-like structure is,
seemingly, more marked than at 5°40'W and, for sure, more marked than in most of the remainder of the western

basin. The marked irregularity between the pink and violet layers originates in fact within the violet layer: the θ-
S diagram is "upward" from the violet-blue layer to the violet one and "downward" from the violet layer to the
pink one, i.e. the diagram displays an irregularity in the violet layer similar to those observed at profiles #2, 4, 5
at 5°40'W in the red layer (Fig.6), all being thus much less smooth than elsewhere in the basin.

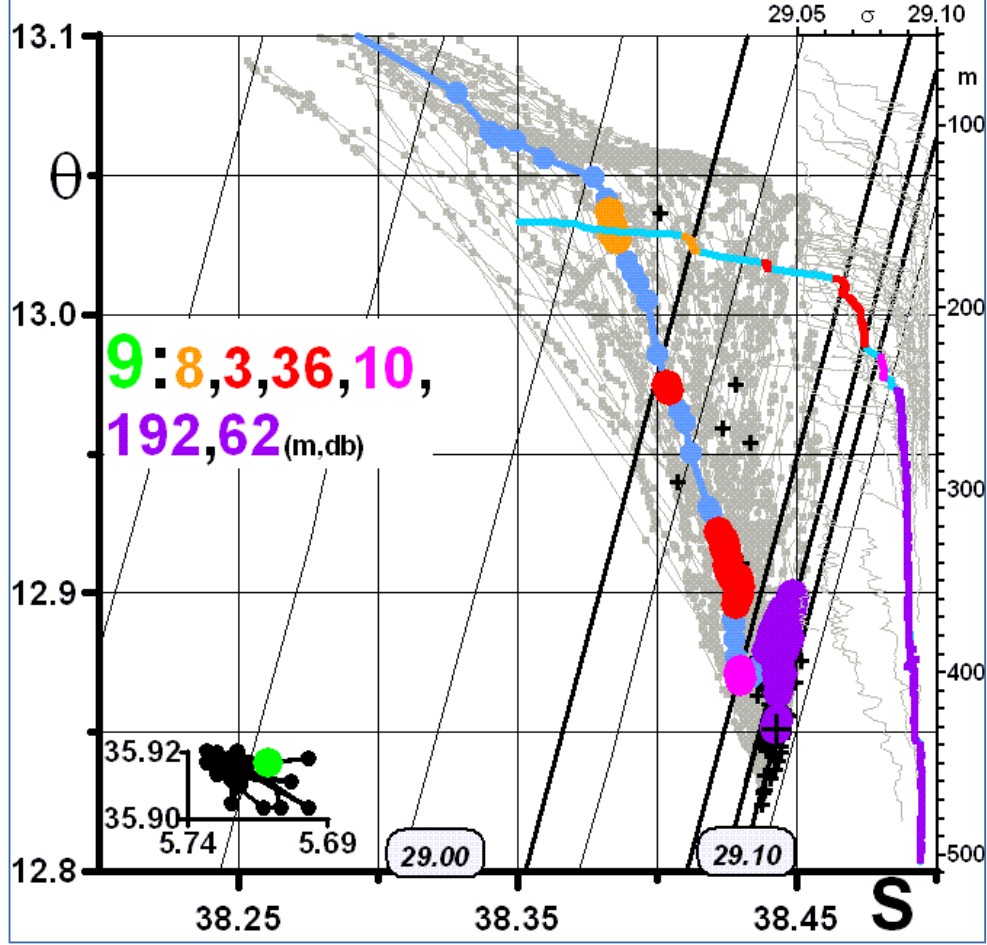

**Figure 9a.** Same as in Fig.6a for profile #9 of the yo-yo time series at 5°43'W. The $\sigma_{max}$ values for all profiles of
the time series are specified with small crosses.




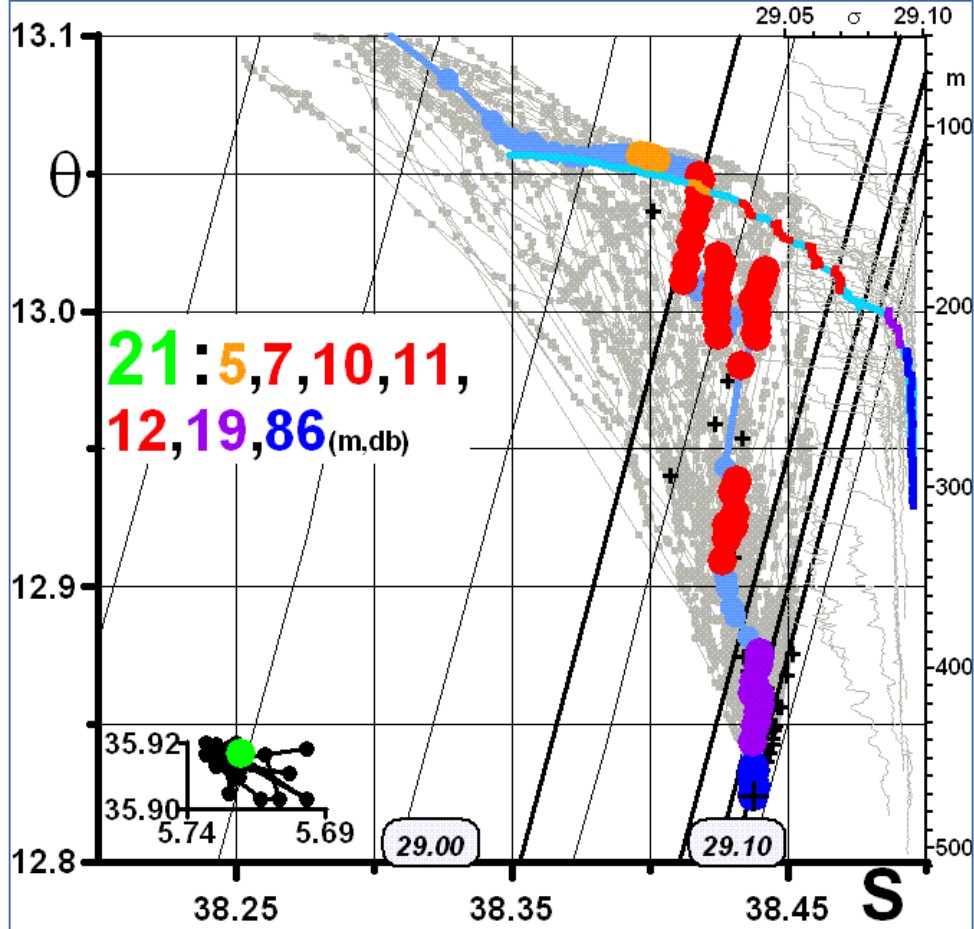

**Figure 9b.** Same as in Fig.9a for #21 at 5°43'W.

The major interest of #21 (Fig.9b) is to show, in the $\sigma_q$ red range (29.080-29.035 kg.m$^{-3}$), several sub-layers
associated with marked changes in the θ-S diagram slope. When just looking at the diagram, it is tempting to
color in pink, a color otherwise missing, the deepest of these sub-layers; but $\sigma_{max}$ =29.0699 kg.m$^{-3}$ of that
sublayer is significantly < 29.080 kg.m$^{-3}$ and the four sub-layers are close to each other and regularly distributed
in both density and depth, so that we associated them with the same red MW. Profile #21 thus provides another

dramatic example (similar to those at 5°40'W) of the mixing processes that drive the transformation of a smooth
profile, that is a part of a θ-S diagram in which the θ-S-$\sigma_\theta$ parameters vary rather slowly and continuously as in
the classical "scorpion tail" part of most θ-S diagrams observed in the remainder of the Sea, into several-step-
like profiles. Note that all these several-thin-step-like profiles of a given color might only be the transitory phase
towards a one-thick-step profile indicating one thick homogeneous layer as observed there (Fig.S1-S49) or even

at 5°40'W (orange on #5).

35    35





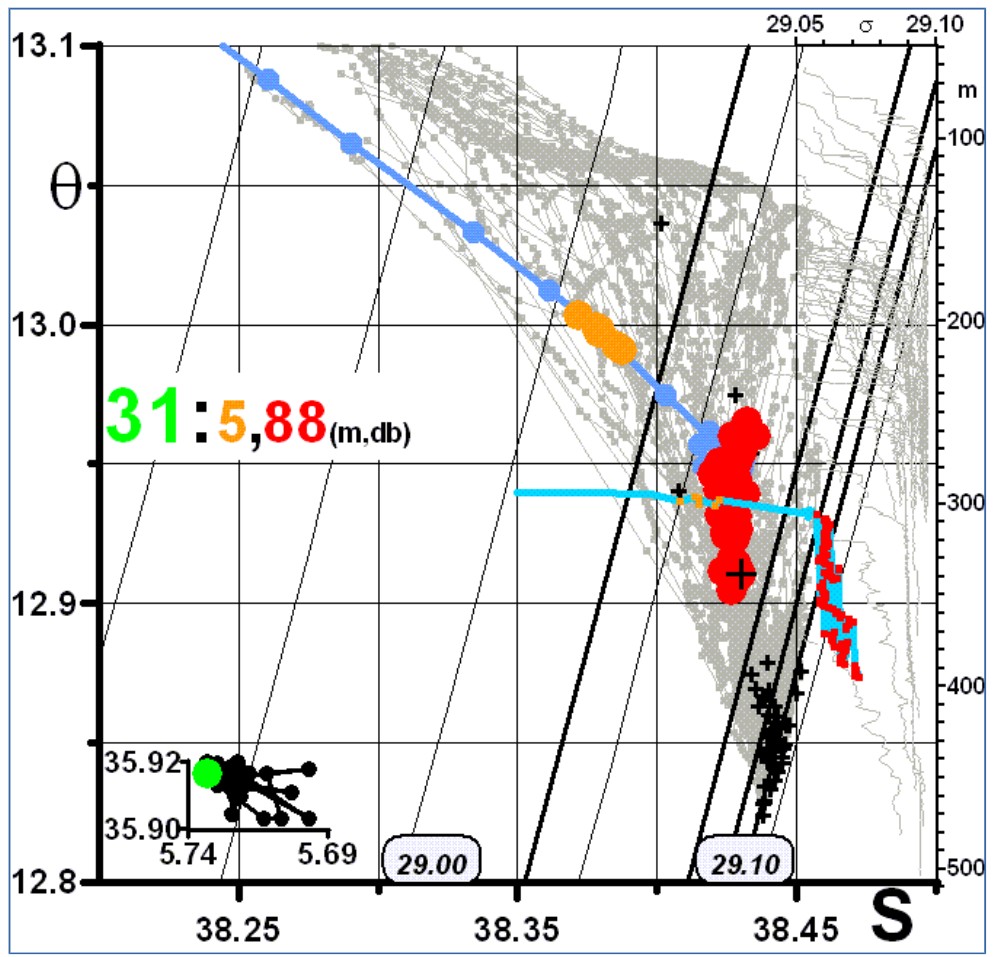

**Figure 9c.** Same as in Fig.9a for #31 at 5°43'W.

Profile #31 (Fig.9c) is both relatively deep (~395 m) and the deepest profile displaying a relatively thick (88 m)

layer of a relatively light (red, $\sigma_{max}$ = 29.0725 kg.m$^{-3}$) MW on the bottom. Marked heterogeneity indicative of intense mixing is evidenced over the whole red layer in which, contrary to most of the previous examples, sub-layers cannot be identified. Note the very sharp and deep AWs-MWs interface in which thin sub-layers of orange water can be identified as well as the absence of the pink (till #34), violet (from #30 to #36) and blue (from #30 to #37) waters.



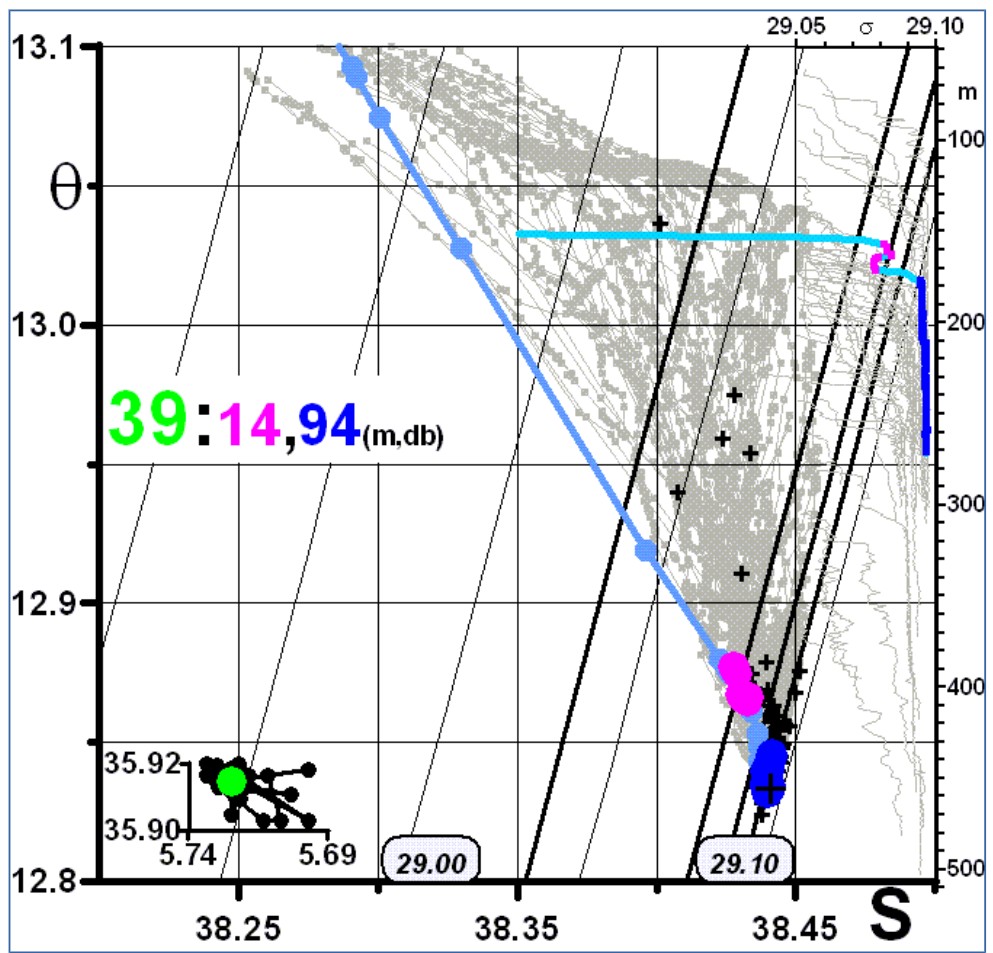

**Figure 9d.** Same as in Fig.9a for #39 at 5°43'W.

Profile #39 (Fig.9d) displays the sharpest AWs-MWs interface (at ~150 m) and the straightest θ-S diagram that is
the most devoid of points in the ranges of interest while linking the AWs with the relatively dense pink MW.
Note that this relatively thin pink layer is markedly unstable, that the violet MW is clearly absent and that the
blue MW can be relatively thick (94 m) over relatively shallow (~270 m) depths.

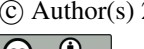



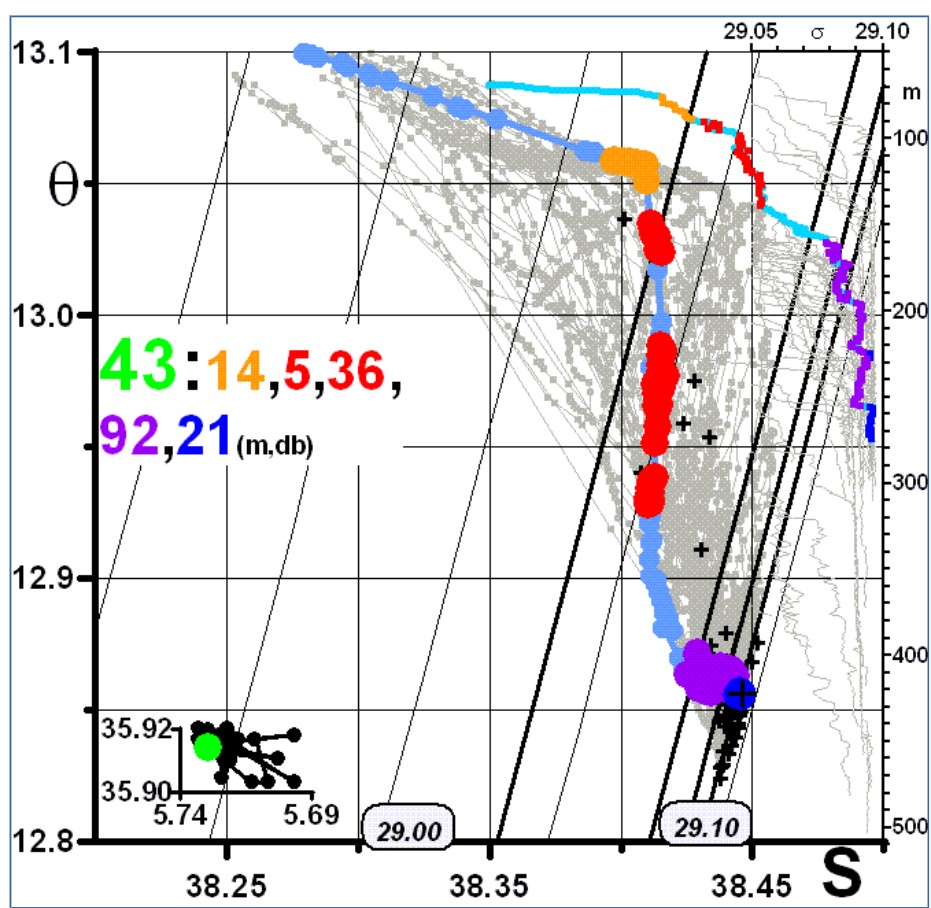

**Figure 9e.** Same as in Fig.9a for #43 at 5°43'W.

Profile #43 (Fig.9e) shows that marked instabilities can also concern a relatively thick (92 m) violet layer that
can be divided into three-four sub-layers sometime characterized by $s_q$ decreasing with depth: in the deepest
sub-layer, the $s_{max}$ is at the top of that layer (~225 m) and it is as large (>29.095 kg.m$^{-3}$) as within the blue layer
~30 m below. Three red sub-layers can also be identified, as well as a thin blue layer, the thickness of which has
markedly reduced since #39 and will continue reducing.


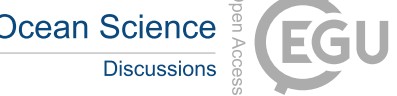

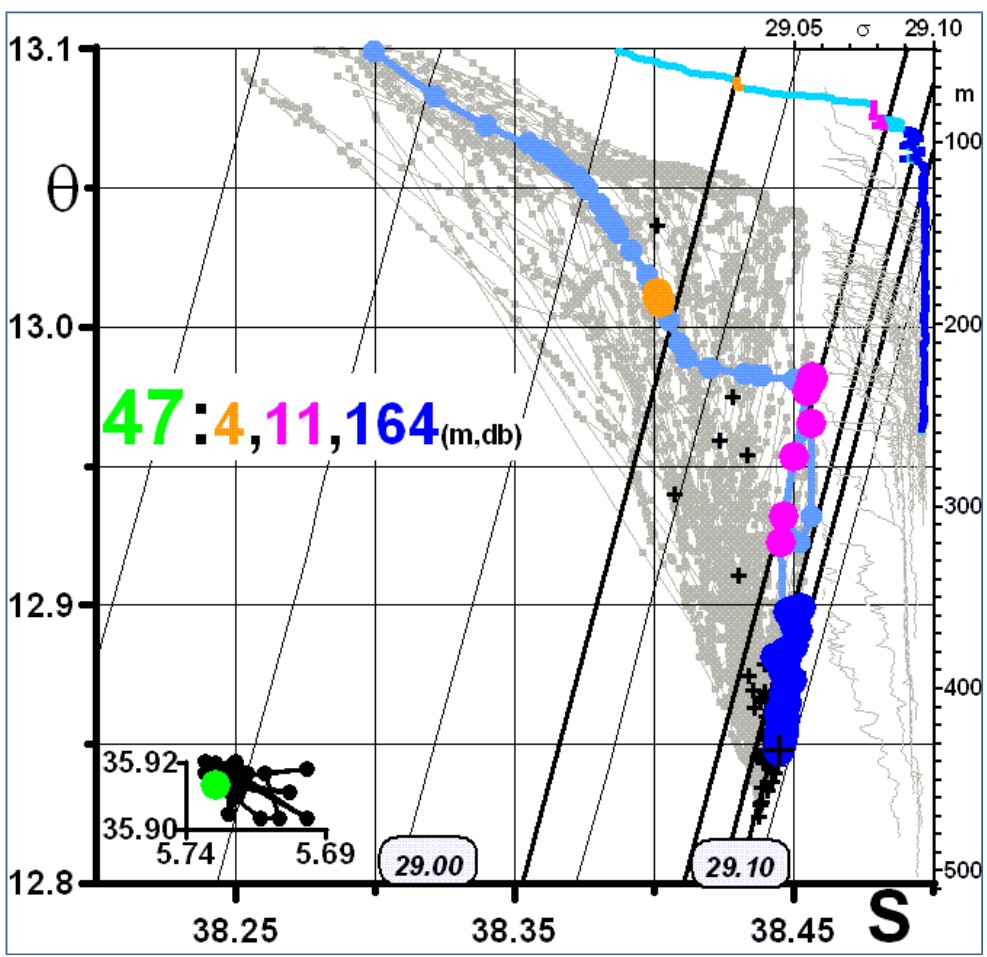

**Figure 9f.** Same as in Fig.6a for #47 at 5°43'W.

Profile #47 (Fig.9f) mainly displays a thick (164 m) and shallow (reaching a depth of ~90 m) blue layer that, even if mainly homogeneous, is concerned by marked instabilities in its top 20 m. Others characteristics are the
absence of any red and violet waters as well as the reduced thicknesses of both the orange and pink ones.





## 6 Discussion

### 6.1 Some generally forgotten evidences


### 6.1.1 All MWs must exit

Any MW is formed, through wintertime dense water formation processes in the north of some given sub-basin, entirely or partially from AWs than have entered that sub-basin. Therefore, at least the part of that MW not

involved in mixing processes leading tot the formation of other MWs there must exit from that sub-basin, hence then from its basin of origin and finally from the Sea. All MWs must thus exit from the Sea, either still identified or mixed with other MWs. This is why, due to the Coriolis effect, waters either inflowing (AWs) or outflowing (MWs) through any passage, in particular the Channel (of Sicily) and the Strait (of Gibraltar), are separated by an interface sloping upwards from Africa to Europe. In any event, depending on how the specific density of a

given MW compares with the densities of the older MWs present in its zone of formation and its surroundings, that MW is qualified as either an intermediate or a deep MW, so that we use for them the general acronyms IWs and DWs.

Qualifying the relatively dense (as compared to the AWs) MWs as either intermediate or deep is not only a

question of vocabulary and/or depth. We have specified our own qualification with arguments that have been synthesized with schematic circulation diagrams basically unchanged from Millot (1987) till Millot and Taupier-Letage (2005a); a conceptual diagram is also proposed for the western basin by Fig.2 of Millot (2009). Considering all other diagrams proposed up to now, we do attribute to the Coriolis effect a role that is much important than in other diagrams, except Nielsen (1912). This is why, as long as waters, hence the MWs in

particular, circulate in a significant way, they circulate in our diagrams along the continental slope at corresponding depths, and with the slope on their right-hand side all around the whole Sea.

Now, we have measured (Millot and Taupier-Letage, 2005b) yearly means of $\sim 10$ cm.s$^{-1}$ (eastwards) at 2700 m on the lower part of the slope off Algeria which, in any event, represents the circulation of a deep MW that is not

only significant but even larger than the circulation of intermediate MWs there as well as elsewhere. Therefore, what mainly differentiates the IWs from the DWs, in particular when they have to cross a major passage such as either the Channel or the Strait that are characterized by sills at $\sim 400$ m and $\sim 300$ m, respectively?

### 6.1.2 The IWs outflow


Some of the MWs approach a given passage while circulating at a depth that will allow them to directly cross this passage, be that depth either lower than the sill depth or sufficiently low to allow, thanks to acceleration and hence thinning (to maintain the fluxes) of that MW in the passage, a direct crossing of it. These MWs are qualified, in a very clear way in the passages at least, as IWs.




Consequently, and due to the Coriolis effect, IWs displaying a significant circulation through the passages are
found mainly on their right-hand side and they are said to outflow. Obviously, a large amount of a given IW will
lead to its presence across a large portion of the passage but its core, characterized by the most typical $q$-$S$-$s_q$
values and/or the largest velocities, will always be found on the right, that is along Europe (not Africa) in both
the Channel and the Strait.

These two passages are actually so narrow, with respect to the E-(P+R) budget that must be equilibrating over
either the eastern basin for the Channel or the whole Sea for the Strait, that the imposed AWs and MWs fluxes
must accelerate through them. Note that, would one basin remain a concentration domain while the other would
become a dilution one (as the Black Sea), a situation with no exchange through the Strait and extreme exchanges
through the Channel can easily be conceived. In any event, acceleration of any water already flowing along a
continental slope leads to an accumulation of that water along the slope, hence to both a rising (of the upper part)
and a deepening (of the lower part) of that water ... as long as allowed by waters above and below. In the
particular case of a set of MWs crossing a passage, and with the most natural assumption that the deeper a MW
the less easy and rapid its crossing of the passage, the shallower MWs will push downwards the deeper MWs.
This will obviously lead the deepest of these MWs to flow away from the right-hand slope, and even on the left-
hand slope of that passage. Now, what is the major difference between the IWs and the DWs?

### 6.1.3 The DWs overflow


Let us imagine a theoretical Sea functioning as a concentration domain but with the dense water formation
process occurring over the whole domain, and not rotating, i.e. without any Coriolis effect, which are not
necessary conditions but allows simplifying what we have to imagine. Or, even more simply, let us fill the
evacuation hole of a bathtub and open the tap; sooner or later, the water in the bathtub will overflow. We do not
see any major difference with the actual DWs formed in the Mediterranean Sea.

More precisely and obviously, any given DW formed in the north of a given sub-basin, hence reaching either the
bottom or an equilibrium depth in that area, must spread out from that area and thus circulate, hence being
submitted to the Coriolis effect. If it reached the bottom, it will also sink as long as the resident MWs it
encounters have lower densities. When it reaches its equilibrium level, be it in between resident MWs or on the
bottom of either its sub-basin or the associated basin, it will continue circulating (as we measured at 2700 m in
the Algerian). But basins being semi-enclosed domains, this DW will sooner or later fill its equilibrium level and
be trapped at that level for a shorter or longer period of time. The only mechanisms able to allow that DW
escaping from that sub-basin or basin are either mixing with other DWs or lifting up by denser and younger
DWs. Mixing obviously occurs always and everywhere and it rubs out the specific signature of that DW. Lifting
up by denser and younger DWs is thus the only mechanism allowing a given DW to overflow, that is outflow
without any definite circulation, through a passage while still keeping recognizable characteristics.



One direct consequence is that DWs sampled in a passage are not necessarily the most recent DWs present in the
associated basin. Assuming a set of DWs are overlying horizontally at the entrance of this passage, the
overflowing DW will just be the less dense of these DWs, be it the DW formed nearby during the past winter of
a DW formed decades ago. This is different from what e.g. Garcia-Lafuente et al. (2007) have said for WDW
when looking for relationships between the WDW formation and its occurrence at Gibraltar. Now, DWs are not
strictly overlying horizontally at the Strait entrance but they are also slightly inclined, as suggested in particular
by the data from the transect at 5°40'W (Fig.6), so that it is in fact the less dense and upper parts of all these
DWs that are overflowing.

Another direct consequence is that, assuming that the filling of a basin by DWs is just ongoing while IWs are
already circulating trough all the passage, the less dense of these DWs will tend to reach the sill of that passage
with a nearly null speed. Since the DWs would then be the deepest and the slowest of all MWs exiting the basin,
they will be pushed by the IWs along the left-hand side of the passage. This obviously occurs even if a basin has
been filled up by DWs for a long time, as long as DWs have overflowing velocities lower than the outflowing
velocities of the slowest IWs. This is why DWs overflow mainly over the African (not European) slope in both
the Channel and the Strait, there possibly up over the Moroccan shelf (Millot, 2009).


In any event, we do not think necessary to invoke the uplifting of e.g. the WDW (in fact all DWs) through the
agency of Bernoulli suction to flow across the Strait (Stommel et al. (1973), Kinder and Parrilla (1987)). To
schematize our own understanding: would the IWs outflow at Gibraltar not only along the whole European
slope, but also over the sill at ~300 m and up to a depth of ~200 m along the African slope, hence roughly over
the northern two thirds of the Strait section "allowed" to the MWs, then the DWs would overflow above ~200 m
along the African slope … just below the entering AWs. This is clearly supported (Millot, 2009) not only by our
reanalysis of numerous GIBEX CTD transects at the Strait entrance but also by the several-year
HYDROCHANGES time series we simultaneously obtained at the sill and on the African shelf (Fig.3): MWs on
the shelf can be significantly denser than MWs at the sill during several-month periods (Fig.22b of Millot, 2009),
clearly indicating that, at least at these times, DWs were overflowing at ~80 m on the African shelf while IWs
were outflowing at ~270 m at the southern Camarinal sill.

### 6.1.4 Generic acronyms must be given to both the IWs and the DWs

Just considering, for the eastern basin, the evidences that all MWs must exit trough the Channel and at least three
IWs are identified in the Ionian up to the Channel entrance (Aegean IW, Cretan IW and Levantine IW, CIESM
group (2001)), and even if LIW is recognized as being the most voluminous one, why continuing to name LIW
the outflow of these IWs from the Channel across the western basin down to the Strait? We already proposed
(Millot, 2013) to name them Eastern IWs (EIW) from the Channel downstream, as long as they can be
differentiated from other MWs.




More coherently, in CIESM group (2001) that must be considered, at least up to now, as the reference document, three DWs (Adriatic DW, Cretan DW, Levantine DW) are differentiated in the eastern basin but no more in the Channel; there, they have been named EOW (the E for Eastern and the O for overflow) or tEMDW or EMDW
(the t for transient and the M for Mediterranean) while, in the Strait, the document recommends using MOW for the whole MO. This document was necessary in the early 2000's even if some basic recommendations are still not considered (as already noticed for WIW), even if we made some confusions in the processes understanding (outflow vs. overflow), even if we maintained unnecessary specifications (such as the M for Mediterranean in e.g. WMDW), and even if some mistakes were made as for TDW (Millot (1999) clearly noticed that TDW was
less dense than WMDW and specified that the D of TDW was for "dense" and not "deep").

This is why we now propose to use, when, where and as long as the original MWs can no more be differentiated, the four generic acronyms WIW, EIW, EDW and WDW, E and the first W standing for Eastern and Western, I and D standing for Intermediate and Deep, the last W standing for Water. These four types of MWs can generally
be identified in the western basin, even if with some misunderstanding of the mixing processes between them (Millot, 2013, 2014a) and thus with obvious inaccuracies in the definition of their interfaces. Following the famous "scorpion-tail image" (Tchernia (1972), reported in e.g. Millot (2013)) and very roughly according to our own definitions of the different groups of MWs present in the western basin, in particular in the western Alboran and at the Strait entrance: WIW is the lightest and it is relatively cool, EIW is relatively warm, EDW is relatively
salty, and WDW is essentially the densest.

**6.1.5 The DWs from the eastern basin must be identified at the Strait entrance**

As first noticed by Sparnocchia et al. (1999; reported in e.g. Millot (1999)), and as described here below with the
above-defined acronyms and our actual understanding of the processes, the relatively different EIW and EDW (Sammari et al., 1999) that outflow and overflow at 100-400 m through the Channel, then cascade down to 200-2000 m. This leads to the circulation and spreading, first in the southern Tyrrhenian and then all around and eventually across the western basin, of relatively warm and salty waters. As specified in Millot (2009) when introducing terms such as upper-TDW/EDW and lower-TDW/EDW, it has been evident for us that, even if EIW
and EDW partially mix together, in particular when cascading from the Channel, there were no reasons for having the totality of EIW (resp. EDW) continuing as an IW (resp. a DW) across the western basin down to the Strait. A priori, it could be that either the lower part of EIW remains temporarily trapped in the western basin or that the upper part of EDW directly outflows through the Strait as other IWs. After some general considerations, we concluded in Millot (2009) that the upper part of EDW probably outflows directly through the Strait, together
with the proper EIW and WIW. In other words and at the Strait entrance, the upper part of EDW outflows while the deeper part of EDW overflows. What are the major differences expected between both parts of EDW?

Even though both have encountered intense mixing processes while cascading from the Channel, the upper part of EDW has circulated just below EIW all around the north of the western basin, which might be a route longer
(in space) than the direct route followed, on average, by the deeper part of EDW between the Channel and the





Strait, but is certainly done at larger speeds in a less erratic way. Therefore, the upper EDW outflowing through the Strait is expected to be younger that the overflowing deeper EDW, hence to have slightly but significantly different characteristics. Also to be noticed is that the deeper EDW present near the Strait entrance will overflow after having been uplifted, either by denser EDW cascading from the Channel or by newly formed WDW. On the

contrary, the upper EDW will always tend to block the uplifting of the deeper EDW, both parts of EDW thus "competing" at the Strait entrance.

In Millot (2009) we somehow assumed that these two parts of EDW (sampled in the mid 1980's) were contemporaneous, hence forgetting to imagine that dramatic changes such as those evidenced in the eastern basin

in the mid 1990's (the Aegean started producing a DW denser than the one produced by the Adriatic, Roether et al. (2007)) could perfectly have occurred in the past. It could thus be possible that these two parts initially resulted from markedly different DWs formed over time (much before the mid 1980's) in the eastern basin; after both have cascaded (simultaneously or one after the other) from the Channel, the denser EDW would spread in the western basin more as a DW while the lighter EDW would spread more as an IW. Note that all these (and

others) long term changes in the hydrological characteristics of the MWs that can be represented by a salinification of ~0.01 per decade have been studied without any consideration of possible long term changes in the AWs characteristics such as the increase of salinity of ~0.05 per year in 2003-2007 (Millot, 2007). Reasons for marked changes in the hydrological characteristics of all MWs are thus numerous and certainly account for the dramatic changes observed in the MO structure and characteristics (see Part 3).


In any event, these above-mentioned evidences have led us to conclude in Millot (2009), using the acronyms we now propose: "At the Strait entrance, we prefer considering a set of IWs (WIW, EIW, upper EDW) and a set of DWs (lower EDW, WDW)". Since 2009 at least, we have thus clearly hypothesized that five types of MWs should be differentiated at the Strait entrance, at least in the mid 1980's.


### 6.1.6 On the importance of differentiating SAW from NACW

As indicated by Fig.1a, the mixing of SAW vs. NACW with a hypothetically homogeneous MO, or more realistically with each of its components, i.e. each of the MWs, leads to mixing lines having markedly different

slopes, hence to markedly different modifications of the original characteristics of the whole set of MWs. Differences are even larger when one considers that the MWs can mix with different parts of the SAW layer, as illustrated by e.g. Fig.2. Now, the analysis herein shows that the $s_{max}$ values inferred from each of the profiles from both the transect at 5°40'W (Fig.6) and the time series at 5°43'W (Fig.9), as displayed by the black crosses on $q$-S diagrams having such a $Dq/DS = 1°C$ scale, are distributed "roughly on the vertical": with such a scale,

MWs are differentiated more on $q$ than on S, and/or the MO is more heterogeneous on $q$ than on S. Just because, essentially, NACW (resp. SAW) has a mean $q$ roughly similar to (resp. markedly larger than) the mean $q$ of the MWs, mixing lines of the set of MWs with the NACW (resp. the SAW) will always be more "horizontal", juxtaposed and differentiated (resp. "vertical", superimposed and confounded). Moreover, considering the $Dq$



(and DS) of each of these AWs, mixing lines with the SAW will thus always be more convergent towards the

MWs, hence will always be more pernicious in suggesting the convergence towards a single q-S point in the

MWs range, hence the occurrence of a homogeneous MO. The differences between the mixing lines of the MWs

with NACW vs. SAW are more completely illustrated in Part 2.

**6.2 How do our previous hypotheses match with the data analysis herein?**

Identification of these five types of MWs with the five groups of θ-S-$\sigma_{max}$ data sets objectively inferred (Fig.4, 5)

from the 49 profiles collected during the yo-yo time series performed within less than 24h at 34°55'N-5°43'W

and that we have colored in blue, violet, pink, red and orange, seems to us almost straightforward.


The blue group is the densest and the associated ranges in density (29.095 < $\sigma_{max}$ < 29.0975 kg.m$^{-3}$), temperature

(12.82 °C < θ < 12.88 °C) and salinity (38.438 < S < 38.452) can obviously be associated with WDW. Note that

the θ and S ranges are relatively large and that, even though some θ values are relatively low, WDW cannot be

characterized by the lowest θ values, as generally assumed up to now; in the same way, WDW can be

characterized by relatively large S values even if not originated from the eastern basin, contrary to what has been

assumed up to now. It seems to us obvious that the large wideness of both ranges must be associated with a

relatively young and not very mixed MW, which could indicate that, at these times, recently formed WDW had

been relatively light and hence able to overflow from the Sea almost rapidly. On the contrary, one can imagine

that an overflowing WDW that would be relatively homogeneous would also be relatively old and would have

been uplifted after a series of severe winter conditions that would have produced a large amount of denser

WDW.

The violet and pink groups can be valuably analyzed together since they display similar distributions while

associated ranges in both θ (~0.035 °C for the violet and ~0.016 °C for the pink) and S (~0.006 for both) are

relatively narrow, hence accounting for a relatively reduced variability indicating relatively mixed MWs. Even

though mean and median θ (resp. S) violet and pink values are larger (resp. lower) than the blue ones, then even

if violet and pink MWs are statistically fresher (by 0.003-0.006) than WDW, we retain that they are markedly

warmer (by 0.01-0.03 °C). Since i) EDW is, by far, the type of MWs that has encountered the largest mixing

process when cascading from the Channel into the Tyrrhenian, ii) characteristic θ and S values are well within

the expected ranges, while this relatively warm type of MW is located just above WDW, iii) we have been

expecting the subdivision of that type of MW in two parts, we think that at least this specific set of yo-yo profiles

has allowed evidencing the two parts of EDW.

Even though the red group is formed by only five sets of θ-S-$\sigma_{max}$ data, it represents MWs that are significantly

warmer and markedly fresher that the denser MWs, EDW in particular. Almost undoubtedly, the red group must

thus be associated with EIW. As compared in particular with EDW, it is relatively heterogeneous, which is





probably due to the fact that it encountered a relatively low mixing while cascading just slightly from the
Channel; one can also consider its natural heterogeneity within the eastern basin (in the Ionian) that could have
been maintained, at least partially, all along its course down to the Strait, as well as the interactions with

overlaying waters of markedly different origin as WIW that, even if not often sampled, must be associated with
the markedly lightest orange group. Note that the red EIW group, defined only by consideration of the sets of
$s_{max}$ values from the yo-yo time series (Fig.5), then appears to occupy the place it (formerly LIW) was expected
to occupy on a q-S diagram in the whole Sea, that is "at the tail of the scorpion" (e.g. Fig.6h, S19, S20, S46).

Finally, let us specify that the relatively low numbers of $s_{max}$ values in the orange and red ranges as compared to
the numbers in the pink, violet and blue ones is mainly due to the specific location of the time series on the upper
part of the southern continental slope at 5°43'W and to the slight (a few %) but significant inclination on the
horizontal of the isopycnals, i.e. the layers of MWs. Proportions would have been reversed with a time series
collected on the upper part of the northern continental slope at exactly the same longitude (see Part 3).


### 6.3 What must be one of the focus of forthcoming studies at the Strait entrance?

Apart from the fact that the MO must now be definitively considered as heterogeneous, not only at the Strait
entrance (east from 5°45'W) but also along the whole Strait (the q-S diagram in Fig.2 is for 6°05'W), important

features illustrated at the Strait entrance are the clear grouping of largest densities (Fig.4a) as well as the marked
layering of the whole set of MWs that are more homogeneous than anywhere else in the western basin (most q-S
diagrams herein). There, such a marked staircase structure occurs in the central part of the Tyrrhenian sub-basin
at depths of 500-2500 m as a consequence of double diffusivity (e.g. Zodiatis and Gasparini, 1996) between the
relatively warm and salty EIW-EDW and WDW. Now, such a structure appears only on both q and S and the

process, also known as salt fingering, leads to density homogenization, hence to a smoothing of $\sigma_q(z)$ profiles.
Double diffusivity has thus consequences opposed to those observed at the Strait entrance.

When we tried to better understand the mixing of a relatively warm and salty IW with the waters above and
below (Millot 2013, 2014b), we looked for in situ examples of our basic computations and focused on profiles

that displayed the classical "scorpion-tail" image, which is a relatively smooth profile displaying a relative q
maximum well above an absolute S maximum. We looked at the GIBEX data set and found several of them at
5°30'W, but none more to the west, in particular at 5°40'W where they were "too strange" and irregular for us.
We started looking for steps in density, computing density gradients only from relatively smooth profiles all over
the Sea and did evidence significant density gradients mainly at 5°30'W.


We do think that "strange" processes are occurring from ~5°40'W, that are increasing up to the Strait entrance
(e.g. at 5°43'W) and lead to q-S diagrams so complex and variable (Fig.6, 9), which display so thick and
relatively homogeneous intermediate layers (e.g. Fig.6d), so large and deep density gradients (e.g. Fig.6e, 9c), so
markedly unstable layers (e.g. Fig.9c, 9e) and so fine a layering as often observed in the WIW layer (e.g. Fig.9b).



Most of these processes do not involve the AWs-MWs mixing and just concern the MWs. And the very specific
feature we have always emphasized there is the acceleration the IWs encounter when outflowing through the
Strait that leads, due to the Coriolis effect, to an intensification of the on-offshore isopycnals slope, not only
between the circulating IWs and the sluggishly moving DWs, but also within the IWs. Computing such a slope is
not an easy task in such a large area where available data are far from being simultaneous and where effects of

the semi-diurnal tide are so large, but estimations from Millot (2009) lead to slopes between the IWs and the
DWs of ~3%, which might be a relatively large value. Indeed, this is roughly the value of the slope of the
southern continental slope within the Strait, which explains why DWs are found in general along that southern
continental slope (roughly at ~200 m), and even over the southern shelf (< 100 m) while mainly IWs are
observed at the southern sill (~300 m).


Therefore, and to try formalizing our ideas with the aim to motivate theoretical analyses and numerical
simulations, could it be that a layer continuously stratified in density when horizontal become discontinuously
stratified when inclined by "just a few %", hence allowing the formation of superimposed sub-layers, as so
clearly evidenced herein? Note that we do not argue herein for the necessity to perform both theoretical analyses

and numerical simulations of the overall dynamics of the MO (we demonstrate in Part 2 and 3 that the MO,
mainly stratified on the vertical at the Strait entrance, rapidly and definitely comes to be stratified on the
horizontal).

**7 Conclusion**

Only partial conclusions can be definitively inferred from the Part 1 of our trilogy that addresses the
heterogeneity of the Mediterranean Outflow (MO) which results from the well-known transformation of Atlantic
Waters (AWs) into a set of Mediterranean Waters (MWs), thanks to dense water formation processes occurring

during the winter in the north of some specific sub-basins of both the eastern and the western basins of the Sea,
all these MWs having to escape from the Sea.

First, a reanalysis of historical (mid-1980's) data available to the whole community since the early 2000's, has
provided definitive evidence about the now indisputable fact that the MO is markedly heterogeneous in the

western side of the Strait of Gibraltar (~6°05'W), hence contradicting what is claimed in particular by Naranjo et
al. (2015) and Garcia-Lafuente et al. (2017). We did not focus herein on the fact that heterogeneity within the
Strait is essentially on the horizontal, which will be done with the analysis of other transects at 6°05'W and
5°50'W in Part 2 and we will show in Part 3 that this intrinsic heterogeneity lead, at the Strait exit, to the splitting
of the MO into veins without any major influence of the bathymetry there.


In the westernmost side of the western basin, we have shown that, even a classical north-south CTD transect
across the Alboran sub-basin (~5°40'W) evidences a significant layering on the vertical of the whole layer of
MWs, with a limited series of relatively homogeneous layers separated by more or less thick interface layers.



More to the west, we have analyzed, for the first time to our knowledge, a CTD yo-yo time series collected
during ~23 h at 35°55'N-5°43'W, that is on the upper part of the southern continental slope at the Strait entrance
(Camarinal sills are at 5°45'W). Even though the transect at 5°40'W confirms this is not the best side of the Strait
for the observation of the IWs, that are defined as the Intermediate MWs outflowing directly through major
passages such as the Strait and the Channel mainly on their right-hand side while the less dense of the DWs (the
Deep MWs) are overflowing through these passage after having been uplifted by denser DWs, five groups of
MWs are clearly identified there. It will be hypothesized in Part 2 and 3 than the DWs would have been even
more clearly identified more to the south.

In particular at 5°43'W, profiles display much more and more marked step-like layers, especially in density, than
anywhere else in the western basin. And for the first time ever, we have provided several example of the
tremendous instability and dramatically intense mixing processes that occur in all these layers at all depths and
that will transform a continuously stratified overall layer of MWs into a set of relatively homogeneous layers
formed by each of the MWs and separated by relatively thin interface layers.

The analysis of all these CTD profiles at the 1-m/db data level, as well as arguments based on our own
understanding of the Sea functioning, lead us to associate these groups with, essentially, the IWs and the DWs
formed in both the Eastern and the Western basins of the Sea, hence leading to the definition of four major types
of MWs: WIW, EIW, EDW and WDW. We provide additional arguments supporting our previous believing
(Millot, 2009) that the EDW could not behave entirely as a DW in the western basin, the upper part of it
probably exiting through the Strait as an IW.


In any event, we note that physical oceanographers are mainly working with a very limited set of in situ
parameters. Indeed, they only measure in situ temperature (T), conductivity (C) and pressure, mainly collecting
data with CTDs either moored or used as profilers, and they can only infer potential temperature ($\theta$), salinity (S)
and potential density anomaly ($\sigma_\theta$) as functions of depth/pressure or time. It is a pity that they did not follow the
valuable example given by e.g. Howe et al. (1974) who efficiently merged chemical and physical data. It will be
a necessity, for in situ physical oceanographers, in particular those working in the Strait of Gibraltar, to associate
with chemists and/or nuclear scientists more closely. Indeed, not considering local sources of pollution for
instance, hence the local introduction of anthropic tracers, it seems obvious that waters originated in either the
eastern or the western basins have different chemical characteristics that must be more considered, furthermore
promising studies are now available (e.g. Roether and Lupton, 2011; Palmieri et al., 2015).

For the time being and only using the presently available instruments and techniques, we just hope that our study
will motivate two kind of actions. At sea, it would be interesting to repeat CTD yo-yo time series at the Strait
entrance (near ~5°43'W) even if only for relatively short daily periods, the longer being obviously the better, at
several locations along the upper parts of both the southern and the northern continental slopes. Ideally, such
locations should be occupied by ships simultaneously performing yo-yo time series, or by autonomous yo-yoing
devices. It appears that the Strait entrance is a place where yo-yo time series can provide especially valuable



information since the MWs are not yet too much mixed with the AWS and they are still essentially superimposed. The Strait entrance is definitively the best place to perform chemical-type measurements, in

particular measurements that need to collect water samples to be analyzed in the lab. Tow-yo cross-Strait CTD transects at the Strait entrance, such as those recommended in Parts 2 and 3 in the Strait itself and at the Strait exit, could provide information much unbiased and reliable than the information provided by classical transects, but care must be taken due to the large tidal variability there.

In the lab, two types of theoretical analyses and/or numerical simulations should be performed. One deals with small-scale mixing processes as specified at the end of the Discussion: could a tilting by a few % of a continuously stratified layer make it becoming a set of superimposed homogeneous sub-layers? Another deals with general circulation: how a set of horizontally stratified waters having to cross a narrow and shallow passage, with some waters outflowing and other overflowing, would organize in and after that passage, hence

checking the validity of the schematic diagram proposed in Millot (2014a) and commented in Part 3?

**Data availability.** Free and easy access of the GIBEX data to everybody via the four MEDAR/MEDATLAS CD's. HYDROCHANGE data available upon request.

**Team list.** C. Millot is retired and works alone.

**Competing Interests.** The author declares he has no conflict of interest.

**Acknowledgments.** *Thanks to the Editor and the referees.* This research did not receive any specific grant from
funding agencies in the public, commercial, or not-for-profit sectors.

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
