# Peer review of "Definitive evidence of the Mediterranean Outflow heterogeneity. Part 1: at the Strait of Gibraltar entrance"

_Ocean Science, 2017_

## Referee Comment (RC1) · Anonymous Referee #1 · 21 Sep 2017

General comments The presence in the Gulf of Cadiz and further downstream, of Mediterranean Water lying at different depths and constituting distinct well identified cores, has been focus of discussion since the years 1970's. Most authors attribute the formation of these veins to bathymetric effects in the Gulf but a few authors sustain the existence of this heterogeneity already in the Strait of Gibraltar. The fact is that the lack of appropriate data has maintained this uncertainty until now. The present manuscript, which is the first part of a sequence of three, is grounded on a set of data collected in the western side (entrance) of the Strait. The main objective is to show that the Mediterranean Outflow is already heterogeneous in the Strait itself. The importance of the manuscript, which is based on detailed CTD data, is not only the evidence of the heterogeneity of the outflow in the Strait but also the suggestions about the way data

collection in the region should be undertaken in the future.

Specific comments In general, the written text is difficult to follow because it is not always clear and straightforward and would benefit from strong simplification/clarification in some places. The figures (maybe too many) illustrate the main conclusions of the manuscript, but some of them could be simplified/clarified, together with the respective captions. Maybe a figure with the bathymetric features of the whole area (entrance, exit and within the Strait) with the respective names, would help the readers not familiarized with the region. Page 11, Fig. 2: the light gray lines of the yo-yo time series are almost invisible; the boxes with information within the figure make it too heavy so, in fact, as most of that information is also in the figure caption, they could be discarded. Line 73: what is the meaning of "northern Ocean"? Lines 142-145: clarify the sentence contained in these lines Lines 295-297: clarify the sentence contained in these lines Line 339: from the observation of Fig. 1, it seems that the mixing lines are converging Line 350-352: clarify the sentence "Just because..." Lines 512-514: clarify the sentence contained in these lines Fig. 6a: confusing figure with theta-S diagrams mixed with the density profiles; does the size of the coloured dots in the theta-S diagrams mean something? The figure caption is also very confusing and does not explain some of the things that appear in the figure (e.g., the green dot, the n3-5, the numbers 23, 10). As the following figures 6b-6f depend on fig. 6a' caption, this caption should be very clear. Line 669-670: clarify the sentence "this periodicity...bottom depth" Line 782: clarify the sentence "These two passages...through them". Lines 798-800: clarify the sentence "let us fill ....Mediterranean Sea" Line 825: explain why "with a nearly null speed" Line 924: explain "a Dtheta/DS = 1oC scale" Line 991: explain the sentence "double diffusivity ...Strait entrance"

Technical corrections In the whole text, there are several cases of wrong letterings (normal instead of symbol) for the potential temperature (q instead of theta) and potential density anomaly (Sq instead of sigma-theta) Abstract: define "entrance" and "exit" of the Strait Line 84: to the Strait, where... Line 106: cut the "Now," Line 118: analyze

in detail Line 157: lower part Line 165: were the only components Page 6, Table 1 legend: a deep water from... Line 206: Gasser et al.2017 is not included in the References Line 247: clarify the definition of the layers (bottom-250 m-200 m-170 m?) Page 11, Fig. 2 caption: there is no reference to the enclosed graphic with lat.-long. values Line 335: The fact that Line 356: a few hours Fig. 3 caption: Since latitudes and longitudes in the figure' axes are in decimal format, there should be a correspondence when lat. or long. values are referred in the caption, e.g., 5o 40'W (- 5.66 oW); HYDROCHANGES CTDs (green) Line 521: ...and red groups in Fig. 5 Line 524: larger than that Line 559: at profile #1 Line 561: a general feature of Line 622: difference from previous Line 639: more to the south or more to the center? Lines 661, 677, 704 and 973: reference is made to Supplementary Information figures which do not seem to be available. Lines 685-688: the "marked irregularity" mentioned in this paragraph is hard to observe in the figure (Fig. 9a) because the coloured dots in the theta-S diagram are too big. Line 745: leading to the Line 756: Millot (1987) is not in the references Line 758: a role that is much more (?) important Line 1038: heterogeneity leads Line 1087: much more unbiased Line 1127: García-Lafuente et al., 2011 does not appear referred in the text Line 1139: Gasser, 2017 is missing (although it is referred in the text) Line 1150: Millot, 1987 is missing (although it is referred in the text) Line 1185: the title of the paper Peliz et al., 2009 is missing

---

## Referee Comment (RC2) · Anonymous Referee #2 · 3 Oct 2017

The paper investigates heterogeneity of the deep and intermediate waters just to the east of the main sills of the Gibraltar Strait. It is one part of a trilogy, and I have not read the two other papers, so in some way, I am poorly placed to evaluate how valuable is the effort and whether it brings original scientific results.

The paper presents an analysis of three sets of T and S (and density) vertical profiles in a semi-qualitative and semi-quantitative way. The profiles that originate from open-access databases (except for the data from Hydrochange sensors that are also invoked) are in themselves interesting and could provide some insights on the processes at plays. Most of the profiles (except for one section) are from the southern part of the western ALboran Sea in the vicinity of the sill. A strong emphasis is given to the variety of properties (T, S, more than stratification) in different classes of potential

density corresponding to a range of identified deep and intermediate source waters, modified by mixing either horizontally or vertically.

I have three main concerns: 1/ I have not really understood how the notion of a layer (in particular 'homogeneous' layer), or of an interface is defined ('by eye' or 'qualitatively' is mentioned). Of course this is really difficult in this system with huge variability in mixing, stirring and internal tides, but some framework would be helpful to reach conclusions (and this is 'done' on figures 2, 6, 9 in some way for defining the layers from the vertical density profile).

2/ I got lost by the mix of detailed description of individual profiles (in particular thickness of different layers), without having a clear sense of what the author wants to demonstrate: is it quantifying the spread of properties (and distribution) in the different density ranges, and whether or not these are stratified or less-stratified layers separated with itnerfaces? Is it to convince that these different water masses all take place in the overflow, and thus how? I suspect this last point might be tackled to some extent in the other two parts of the trilogy, but it seems to me that some information on flow/transport is also required that is missing here. If it is more the first objective, clearly more effort should be devoted to synthesize the results. To elaborate a little bit on that, I found figures 4 and 5 fairly synthetic ('the yoyo time series', although I missed there a plot of the layer thickness distributions). Actually, why is this presentation not also continued with the other two data sets?

On the other hand, figures 2,6, 9 which contain a wealth of information, are much too detailed and their presentation lost me. Figure 7 could be considered a synthesis of figure 6. It is however a little difficult to interpret. Figure 8 summarizes the yoyo time series. If I understand it right, most of the deep layer is nearly unstratified (in density), with a very small portion of the water column in interfaces (this could be quantified). It would also be nice to mention how the tidal currents at this site evole in this deep layer (when is there eastward or westward flow?)

3/ the structure and contents are not what is expected of a scientific paper. There are lots of judging comments on other published work or general statements which dont have their place in the manuscript (starting in the introduction). The presentation starts with an interesting figure 1, which is somehwat followed during 2, but not really afterwards (so I wonder how important it is for the paper), until mentioned again in section 6.1.6. The presentation of individual profiles is spred over many pages and with figures 2, 6 and 9. I am wondering whether they could not be summarized in a joint section. Then, the discussion starts with a very long 6.1 entitled: 'Some generally forgotten evidences'. Frankly, I dont think that the assumption should be that we 'readers' have forgotten some rather general considerations on the western Med ocean circulation and processes taking place near Gibraltar Strait. As I stated, maybe 6.1.6 could be summarized and kept. 6.3 deals with what should be done next. I dont think that this has a place in a discussion section. This discussion of what could be done next is indeed continued in part of the conclusion section, which might not be the right place for it.

Alltogether, I feel that the paper should be strongly streamlined (at least, cut by half), and edited to avoid statements that can be interepreted as judgmental or oversimplifying, or not directly related to the work presented. Some of the profile figures could be kept in supplementary material, with a short descriptive presentation of a couple of typical examples and more effort to dscribe synthetically the variety of profile characteristics.

One question: the ranges of density given for the different watermasses is rather narrow: how is it taking care of the low frequency (multidecadal) evolution of deep and intermediate Mediterranean water masses.

---

## Author Comment (AC1) · 12 Nov 2017

Dear Referee #1,

I lengthly prepared my answers to your very helpful comments and found convenient using italic, bold and underlined characters so that I preferred uploading my answers as a "pdf supplement".

I hope you will accept answering my questions, in particular those that will help improving my paper.

Many thanks for your help.

Please also note the supplement to this comment:

https://www.ocean-sci-discuss.net/os-2017-52/os-2017-52-AC1-supplement.pdf

**Supplement:**

**ANSWERS TO ANONYMOUS REFEREE #1**
----------

First of all, **I sincerely and warmly thank you for your careful reading of my paper and your very helpful comments**.

This being said, and even if you do not want to review a revised version of the paper, **I would appreciate receiving, either officially through the OS website or personally, with the help of the OS secretary, which will respect your anonymity and allow a more frankly discussion, your answers to my questions and your comments to the arguments I present.**

*General comments The presence in the Gulf of Cadiz and further downstream, of Mediterranean Water lying at different depths and constituting distinct well identified cores, has been focus of discussion since the years 1970's. Most authors attribute the formation of these veins to bathymetric effects in the Gulf but a few authors sustain the existence of this heterogeneity already in the Strait of Gibraltar.* Even though I recently published several papers about the Strait, I do not consider myself as a specialist of it. I am just a scientist who could be considered as a specialist of the Sea and has some specific ideas about the Strait functioning. As for "*a few authors sustain the existence of this heterogeneity already in the Strait of Gibraltar",* hence for authors who somehow share/d some of my major hypotheses, I only know the relatively old papers by Howe (1974, 1982). **Please, so as to help me presenting an as complete as possible Background Sect., please could you list who are the "*few authors*" sharing my hypotheses?**

*The fact is that the lack of appropriate data has maintained this uncertainty until now.*
**I do not share this point of view since there is no "*lack of appropriate data".*** Indeed, I analyze data collected in 1985-1986 (mainly in Parts 1 and 2 but also in Part 3) that are available to everybody for free since 2002, and data collected in 2009 (in Part 3) that are probably available to anybody, as myself, who did not participate in the experiment but is willing to collaborate with the participants. Even though these data sets are not ideally appropriate, the fact is that either they have never been analyzed at all (for the GIBEX yo-yo time series, at least as far as I know) or they have been analyzed with the postulate (sic) that the MO was homogeneous at the Strait entrance and within the Strait itself. **Since I do not present and use new data sets, the "*uncertainty until now*" is only due to the "*lack of appropriate* analyzes".**

*The present manuscript, which is the first part of a sequence of three, is grounded on a set of data collected in the western side (entrance) of the Strait.*
It is the eastern side.

*The main objective is to show that the Mediterranean Outflow is already heterogeneous in the Strait itself.*
This is right and due to the fact that the most recent (2015 and 2017) papers about the Strait claim that the MO is homogeneous in the Strait itself while (only recently) agreeing that the MO is heterogeneous at the Strait entrance.

*The importance of the manuscript, which is based on detailed CTD data, is not only the evidence of the heterogeneity of the outflow in the Strait but also the suggestions about the way data in the region should be undertaken in the future.*

I understand this general comment as "the importance of the manuscript is significant".

*Specific comments In general, the written text is difficult to follow because it is not always clear and straightforward and would benefit from strong simplification/clarification in some places. The figures (maybe too many) illustrate the main conclusions of the manuscript, but some of them could be simplified/clarified, together with the respective captions. Maybe a figure with the bathymetric features of the whole area (entrance, exit and within the Strait) with the respective names, would help the readers not familiarized with the region.*

I am obviously ready to clarify and simplify the writing as much as possible and even move more figures to "Supplementary Information", and **I will strictly follow the Editor's recommendations**.

*Page 11, Fig. 2: the light gray lines of the yo-yo time series are almost invisible; the boxes with information within the figure make it too heavy so, in fact, as most of that information is also in the figure caption, they could be discarded.*

I "discover" that colors on the screen of a PC are not exactly retrieved in a printed version of a figure and I will modify the original colors accordingly. As for the information boxes, I personally consider that a self-explanatory figure, as long as scientific information is not masked, allows easily and efficiently comparing similar figures, as long as the caption has been read once, and also allows projection of the figure on a screen for a large audience. In any case, **I will follow the Editor's recommendations**.

*Line 73: what is the meaning of "northern Ocean"?*

The first sentence of the Introduction (l. 66-69) introduces the names "Mediterranean Sea", "Atlantic Ocean" and "Strait of Gibraltar". The second sentence (l. 69-71) only uses "Sea", and the word "Strait" is used at the beginning of the third sentence (l. 71); I thus thought/think that the term "northern Ocean" (l. 73) can only be linked to the Atlantic. Note that the "specific use of the terms Ocean, Sea, basin, sub-basin, Strait and Channel we make" is specified in the 1$^{st}$ sentence of the Background Sect. 2.

*Lines 142-145: clarify the sentence contained in these lines*

The major controversy I address is about homogeneity vs. heterogeneity of the MO. Addressing this major controversy is made difficult by the fact that there is an underlying and relatively minor controversy about what does heterogeneity mean, furthermore arguments claiming for homogeneity have markedly evolved in recent years. Indeed, while I have always claimed for the occurrence of up to five identifiable MWs at the Strait entrance, what is now strongly supported by the yo-yo time series data set (without any specific analysis), most other papers that have claimed (up to 2015) for the occurrence of only two MWs (LIW and WMDW), now (2015-2017) tend to recognize four of them. Furthermore we do not identify the various MWs in the same way, we do not agree on which MWs could be possibly identified at the Strait entrance, hence on what does heterogeneity mean.

*Lines 295-297: clarify the sentence contained in these lines*

In propositional logic, this is what is known as transposition. You can simply have a look at https://en.wikipedia.org/wiki/Transposition_(logic).

*Line 339: from the observation of Fig. 1, it seems that the mixing lines are converging*
Fig.1 does not represent observations but conceptual mixing lines that do converge in $\theta$, S (and $\sigma_\theta$) ranges (in fact a single point) possibly associated with a given MW. Observations are shown in Fig.2 and associated mixing lines do not converge in realistic MWs ranges: therefore, one cannot think about a "single given MW", hence about "a homogeneous MO".

*Line 350-352: clarify the sentence "Just because. . ."*
Relatively homogeneous components of the MO are identified by different sets of dots in cyan. These sets (in particular at points 6, 3, etc.) cannot be associated with unmixed MWs that are markedly cooler and saltier (e.g. Fig.4a). On must thus invoke a re-homogenization process of waters that have previously (in the Camarinal sills surroundings … but not yet at 5°43'W) resulted from a mixing of MWs with AWs. Such a process is briefly introduced at the end of Sec. 3.1 and illustrated with more details in Part 2.

*Lines 512-514: clarify the sentence contained in these lines*
As illustrated by Fig.1, the mixing of two given waters (e.g. an actual NACW and a hypothetical MO) characterized by two different points on a $\theta$-S diagram is characterized by points located in between, i.e. on the mixing line joining these two specific points. It is thus clear (from Fig.4a and 5) that the blue water cannot result directly from the mixing of the pink and violet ones. This could be possible only assuming that, where the blue water was formed, either the pink or the violet waters (assumed to be present in the zone of formation) would have been the coolest and saltiest (i.e. cooler and saltier that the actual blue values), and then would have encountered mixing (with warmer and fresher waters) more intense than the blue water itself. The blue MW that is the densest, the coolest, the saltiest (from Fig.4a, 5a, 5b) of the MWs while being possibly the shallowest (Fig. 5c) is also the youngest (less homogeneous / most $\theta$-S spread one, Fig. 4a), hence the one formed in the nearest zone of formation (necessarily the north of the western basin). **The actual fact that the densest of the MWs has been observed to be the shallowest can only be a direct consequence of its southernmost location due to a north-south tilting of the MWs, which supports my schematic diagram and understanding of the processes presented in Sect. 4.6 and Fig.19 of my Part 3 paper.**

*Fig. 6a: confusing figure with theta-S diagrams mixed with the density profiles; does the size of the coloured dots in the theta-S diagrams mean something? The figure caption is also very confusing and does not explain some of the things that appear in the figure (e.g., the green dot, the n3-5, the numbers 23, 10). As the following figures 6b-6f depend on fig. 6a' caption, this caption should be very clear.*
**I agree that the caption could have been clearer.** However:
-$\theta$-S diagrams are not "mixed" with density profiles. As indicated in the caption, $\theta$-S diagrams are presented together with two inserts: one (in the lower left corner of the diagram) presents the latitude-longitude position (near 5°40'W=5.66°W), the other (in the upper right corner of the diagram) presents the $\sigma_\theta(z)$ profiles.
-as for the colors, it is indicated in the caption that coloring is first inferred from the $\sigma_\theta(z)$ profiles in which homogeneous layers are easily identified by relatively constant $\sigma$ values (vertical portions of the

$\sigma_\theta(z)$ curves); as indicated, specifying the color is then made according to the separating isopycnals (specified only on the θ-S diagram). Colored dots are plotted with a normal size on the $\sigma_\theta(z)$ profiles not to mask the small scale features that characterize these relatively homogeneous layers. They are plotted with a larger size on the θ-S diagrams to smooth these small scale features and somehow give a better and more realistic idea of these homogeneities.

-it seemed obvious to me that the profile number specified in the caption and retrieved in green within the figure (#1 for Fig. 6a) should be directly associated with the position of the green dot, furthermore I specified (l. 358) that profiles on all cross-Strait transect are numbered from south to north.

-it also seemed obvious to me, furthermore I specified the units (m/db with a specific comment in l. 321-324), that the colored numbers would be directly associated with the thicknesses of the various homogeneous layers (which can be roughly but easily checked from the σ(z) profiles), and that n3-5 (with 3 and 5 in orange and n in black) would directly be understood as "several layers 3-5 m/db thick of orange water").

In any case, **I will make the Fig. 6a caption clearer**.

*Line 669-670: clarify the sentence "this periodicity. . .nominal bottom depth"*

The nominal bottom depth has already been defined several times (l. 397, l. 485, l. 533, l. 662) to be 250-300 m and it is specified (l. 398, l. 667) that the yo-yo time series lasted ~23 h. The sentence specifies that "a semi-diurnal periodicity appears from the maximum and minimum thicknesses of the MWs overall layer above the nominal bottom depth", which means that, "the thickness of the MWs altogether above 250-300 m displays maxima (near #16-22 and #42-48) and minima (near #4-6 and #27-33) that are roughly 12-13 h apart". In my answers to the comments from Referee #2, I specify that the layer of MWs (resp. AWs) can be qualified as lower (resp. upper), just to avoid using "deep" that I reserve to the differentiation with "intermediate", the former (resp. latter) MWs circulating sluggishly in the interior (resp. significantly along the continental slope) in the Sea and overflowing (resp. outflowing) through the Strait. Also see below my answer to your comment about l. 825.

*Line 782: clarify the sentence "These two passages. . .through them".*

Such a sentence results from basic computations involving, over a given domain, the budget E-(P+R) and representative parameters (as salinity) of the various inflowing and outflowing waters. For the Sea and the Strait, the budget (yearly loss of ~1 m of freshwater) represents the difference (~0.1 Sv) between the AWs and MWs fluxes; and considering typical values for e.g. S as in Fig.1 leads to fluxes in the order of ~1 Sv. In the case of the AWs within the Sea, i.e. without any topographic constrain, they circulate anticlockwise (due to the Coriolis effect) as alongslope currents several 10s km wide; to maintain the fluxes, AWs must thus accelerate through the narrow (~10-km wide) Strait, which will increase in particular the slope of the AWs-MWs interface there. I make similar comments for the intermediate vs. deep MWs: circulating MWs (the IWs) must increase their speed when they outflow through the Strait, so that the IWs-DWs slope must increase, which rejects the overflowing DWs more to the south.

*Lines 798-800: clarify the sentence "let us fill . . ..Mediterranean Sea"*

In the Background Sect. 2 (l. 172-175), I specify that "I never thought necessary to invoke the uplifting of deep MWs through the agency of a suction effect" as invoked in all other papers since decades (see the references I give). I always thought that DWs overflow from the Sea (or more precisely tend to flow out from the Sea without considering the IWs) just as water overflows from a bathtub having a

filled evacuation hole and an open tap.

*Line 825: explain why "with a nearly null speed"*
I consider that IWs are those that are continuously circulating, first alongslope anticlockwise within the Sea, then through a passage along its right-hand side where they are said (by me at least) to outflow. And I consider that DWs are just sluggishly circulating within the Sea so that they have a nearly null speed everywhere in the Sea, exactly as the bath water in the bathtub example. The DWs get a significant speed just when they flow out or overflow (according to my definitions at least), i.e. when they reach the sill, exactly as when the bath water flows over the edge of the bathtub.

*Line 924: explain "a Dtheta/DS = 1oC scale"*
$\theta$-S diagrams are necessarily plotted over some specific ranges of both $\theta$ and S with $\theta$ in °C and S dimensionless (according to the "practical salinity scale 1978"). The ratio of both ranges $\Delta\theta/\Delta S$ thus defines a scale in °C. Slopes in $\theta$-S diagrams can be compared only if scales have the same numerical value (1 in our case).

*Line 991: explain the sentence "double diffusivity . . .Strait entrance"*
Double diffusivity is characterized by a layering of only $\theta$ and S that leads to density ($\sigma_\theta$) homogenization, hence to a smoothing of the $\sigma_\theta(z)$ profiles through some MWs (you certainly know that $\sigma_\theta$ is used for "potential density anomaly"). Such a well-known process has thus consequences opposed to those observed at the Strait entrance where some unknown process leads to an increased layering of the $\sigma_\theta(z)$ profiles through some MWs. To help specifying what could be such a process, I hypothesize that a layering of these MWs at 5°43'W larger than more upstream (i.e. at 5°40'W herein and even more to the East, as shown in my 2013 and 2014b papers about LIW) could be due to a tilting of the MWs linked to the Coriolis effect and to the necessary acceleration of the MO in the Camarinal surroundings.

*Technical corrections In the whole text, there are several cases of wrong letterings (normal instead of symbol) for the potential temperature (q instead of theta) and potential density anomaly (Sq instead of sigma-theta)*
**I am sorry but I did not check enough the conversion of my docx files into pdf ones and I did not realize that errors occurred in converting the Symbol format only in the end of my files (in this paper after l. 289-302 only) and in a very strange way, for instance on l. 553 and not on l. 554! I will obviously check the totality of my files in the revised versions.**

*Abstract: define "entrance" and "exit" of the Strait*
I am dealing with the Mediterranean Outflow and thus imagined that "entrance" and "exit" of the Strait would be directly associated to its "eastern" and "western" sides.

*Line 84: to the Strait, where. . .*
I am not sure I correctly understand the comment. The previous and first paragraph of the Introduction ends with a sentence (l. 77-81) explaining that "homogeneity" of the MO has generally been a

"supposition" which has never been tested so that I qualify it as a "postulate". Meanwhile, I have always claimed that several types of MWs can be continuously identified. Seems to me that these two sentence conveniently introduce the major controversy (homogeneity vs. heterogeneity of the MO) I address.

*Line 106: cut the "Now,"*
I would have appreciated having an argument but I will replace "Now" by "This being said".

*Line 118: analyze in detail*
I will remove the "s".

*Line 157: lower part*
I am not sure since I could have written "an upper part and a lower part". In French, it would be "les parties supérieure et inférieure", then with an "s" at "parties". I will ask English-fluent persons.

*Line 165: were the only components*
Yes, were.

*Page 6, Table 1 legend: a deep water from. . .*
Same answer as for the comment about l. 157.

*Line 206: Gasser et al.2017 is not included in the References*
I only had a manuscript and now have the exact reference of this now-published paper.

*Line 247: clarify the definition of the layers (bottom-250 m-200 m-170 m?)*
I imagined that giving four depths would allow easily understanding that the three layers were defined by "bottom to 250 m", "250 m to 200 m", "200 m to 170 m". I will modify my writing.

*Page 11, Fig. 2 caption: there is no reference to the enclosed graphic with lat.-long. values*
I will refer to the inclosed graphic.

*Line 335: The fact that*
There are two different facts: one is "a relatively coarse sampling", another is a "not north enough transect". I think that the correct writing is "The facts that ..." but I will ask.

*Line 356: a few hours*
I will modify.

*Fig. 3 caption: Since latitudes and longitudes in the figure' axes are in decimal format, there should be a correspondence when lat. or long. values are referred in the caption, e.g., 5o 40'W (- 5.66 oW);*
I agree. Just note that I am doing all figures by myself and I don't know how to write automatically values in the degree-minute format. Even though correspondences such as between 40' and 0.66° is almost straightforward, I will add the correspondences.

*HYDROCHANGES CTDs (green)*
Yes.

*Line 521: . . .and red groups in Fig. 5*
I can add "in Fig.5" but I must note that these words would also have had to be added in the previous paragraph dealing with the blue group. Also note that these two paragraphs come just after a paragraph dealing with Fig.5a (l. 482-490) and another paragraph dealing with Fig.5b and Fig.5c (l. 499-505) so that links between both sets of paragraphs might be obvious. Finally, note that comments about the groups characteristics are illustrated in all other figures of the paper.

*Line 524: larger than that*
I am not sure since I want to say "... is larger than the spreading associated with the pink group, the spreading associated with the violet group and the spreading associated with the blue group". All three spreadings, furthermore they are inferred from 7, 20 and 16 profiles, respectively, are different.

*Line 559: at profile #1*
In case a word must be added before the sign "#", it should be "point" since I used "at". I could also have written "from (profile)". **I will ask the Editor for a more general writing: "Does a word must always be specified before the sign #"?**
**I will also ask the Editor about the possibility to use "m" in a sentence (as in l. 560 and in reference to your comment about my use of "h" in l. 356).**

*Line 561: a general feature of*
Yes.

*Line 622: difference from previous*
I don't know and will ask.

*Line 639: more to the south or more to the center?*
I understand your comment that, seems to me, is "influenced" by the fact that the central part of the transect is much deeper than its southern part: there will always be more DW in the central part of any north-south transect than in its southern part. I deal with "north" at the beginning of the sentence (l. 638) and want to introduce the notion of a "north-south sloping" that is illustrated in particular by Fig.7a (more specifically by the kriging plot; see what I specified to Referee #2 in this respect). In addition, and as demonstrated in all my papers, I am convinced that the densest MW (blue herein) is the one found in the southernmost part of any transect, whatever the bottom depth is.

*Lines 661, 677, 704 and 973: reference is made to Supplementary Information figures which do not seem to be available.*
You should have asked the OS office … furthermore a lot of interesting information is contained in what is named "Supplement" when I access "My Manuscript Archives" where I have several sections: MS Records, Interactive Discussion, Minor Revision, Initial Submission. In this last section, and at the end of the "Topic Editor Initial Decision" (before my answer to the Editor), you have "File Upload (10 Jun 2017) … "Supplement". I easily got my Supplementary Information file.

*Lines 685-688: the "marked irregularity" mentioned in this paragraph is hard to observe in the figure (Fig. 9a) because the coloured dots in the theta-S diagram are too big.*
I agree. I previously explained (about Fig.6a) that I choose to have relatively big dots for colored parts of the $\theta$-S diagrams. I will add small black dots over the big violet ones.

*Line 745: leading to the*
Yes.

*Line 756: Millot (1987) is not in the references*
Yes, I will add it.

*Line 758: a role that is much more (?) important*
Yes. I wanted to say "more important" and I will correct.

*Line 1038: heterogeneity leads*
Yes.

*Line 1087: much more unbiased*
Same answer as for l. 758.

*Line 1127: García-Lafuente et al., 2011 does not appear referred in the text*
Yes, I will remove.

*Line 1139: Gasser, 2017 is missing (although it is referred in the text).*
Answer as for l. 206.

*Line 1150: Millot, 1987 is missing (although it is referred in the text)*
Answer as for l. 756.

*Line 1185: the title of the paper Peliz et al., 2009 is missing*
Yes, I will add it.

---

## Author Comment (AC2) · 12 Nov 2017

Dear Referee #2,

I lengthly prepared my answers to your very helpful comments and found convenient using italic, bold and underlined characters so that I preferred uploading my answers as a "pdf supplement".

I hope you will accept answering my questions, in particular those that will help improving my paper.

Many thanks for your help.

Please also note the supplement to this comment:

[Figure]

https://www.ocean-sci-discuss.net/os-2017-52/os-2017-52-AC2-supplement.pdf

**Supplement:**

**ANSWERS TO ANONYMOUS REFEREE #2**
* * *
First of all, **I sincerely and warmly thank you for your careful reading of my paper and your very helpful comments**.

With this, and even though you qualified, in particular, the **scientific significance as poor** and the **scientific quality as fair**, I noticed that **you would like to review the revised paper**. I consider that this is an indication of **your open-mindedness** so that **I hope you will answer the questions I ask you**. **In case you cannot or do not want to answer "officially" on the OS site, I am sure you can answer at least to me personally, with just a few-lines overall comment to the Editor, via any person in the OS office, which will respect your anonymity and allow a more frankly discussion between us.**

**Please, consider that the length of my overall answer is proportional to my willingness to provide you and the Editor with explanations as clear as possible, and that the use I make of underlying, letter sizing and coloring (essentially with ? and !) just aims at helping you and the Editor keeping in mind the major points of my answer.**

*The paper investigates heterogeneity of the deep and intermediate waters just to the east of the main sills of the Gibraltar Strait. It is one part of a trilogy, and I have not read the two other papers, so in some way, I am poorly placed to evaluate how valuable is the effort and whether it brings original scientific results.*

- Whatever your personal wishes and/or constrains that prevented you from accepting to review the two other papers are, **I am sorry to say that you did not catch what I think is a tremendously valuable interest of this Part 1 paper, independently of my own analyses!** Indeed, I would have been pleased to read some detailed comments about **Figures 4a and 5a,b,c that are just a rough and fully objective presentation of raw data**, i.e. of the maximum density values associated with a yo-yo time series of CTD profiles collected within ~23 h at the same nominal location; note that, on these two sets of figures, colors are redundant with separation isopycnals, and that both can possibly be omitted.
- **When considering in particular the pink, violet and blue points, don't you think that their relatively homogeneous distributions in density (Fig. 4a), temperature (Fig. 5b) and salinity (Fig. 5c) evidence, for the first time ever, the fact that the MO entering the Strait is definitively layered, being composed of a set of homogeneous components/MWs that can be distributed over relatively large depth ranges (Fig. 5a)?**

*The paper presents an analysis of three sets of T and S (and density) vertical pro- files in a semi-qualitative and semi-quantitative way.*

- **I disagree with the comment** that my paper "*presents an analysis*". First of all, my paper "essentially presents a set of raw data". Please, consider Fig. 4a, 5a, 5b and 5c without any color and any separation isopycnal, i.e. a set of data presented in a way as rough as possible: **could you let me know what would you personally infer from such black-and-white bare plots of data?**
- **I do not present** "*sets of T and S (and density)*" profiles. Please, carefully re-read at least my Abstract. In its first part, I deal with homogeneity vs. heterogeneity of the MO and with its

splitting into veins, which can only be a dynamical (i.e. σ-related, not θ or S -related) problem; in its second part, I deal with "density instabilities", "isopycnals slope", "the Coriolis effect", and "velocities of the MWs". **Do you agree that my major concern is thus with density (σ), not with T/θ and S?**

- In addition, Figures 4a and 5b,c clearly demonstrate that, contrary to what has been thought up to now, not only in the Strait but also in the Sea, MWs cannot be identified by some specific ranges in either θ or S. **Do you agree that, at least at the Strait entrance, MWs can only be identified by their σ?** Note that the specific situation of having two MWs characterized by two sets of data in different θ-S ranges leading to the same σ cannot be practically encountered since such MWs would easily mix. This is what occurs for instance with the different IWs in the eastern basin at the entrance of the Channel (of Sicily) which, in the western basin, prevents from identifying LIW in particular (differentiating it from CIW and AIW) and supports my requirement (Sect. 6.1.4) for generic acronyms, EIW in this case.

- As for what concerns "*vertical profiles*", **I deplore that you did not emphasize the tremendous interest of yo-yo time series**, in particular in a place such as the Strait entrance where the internal tide has so dramatic consequences. Obviously, the yo-yo time series I am presenting is unique (to my knowledge) and such a strategy has to be more frequently / systematically used to check and demonstrate its specific value and, would you have accepted / been able to review Parts 2 and 3, you would have found numerous examples of their significance and interest. At least for what concerns the Strait dynamics, and after having had a look just at Fig.2j in Part 2 that is self-explanatory, **would you recommend performing more yo-yo time series even if, ship-time being limited, to the detriment of single isolated vertical profiles along transects?**

- I do not understand what does "*analysis … in a semi-qualitative and semi-quantitative way*" mean. The only way a sea experimentalist has to analyze data is, often, to describe them from a qualitative point of view and, in any case, this is what she/he must do at first. This being said, I have always tried to perform quantitative analyses, I mean analyses involving computations, which justifies the comment I make (l. 382-387) about my willingness to collect time series of current (all my campaigns within the Sea as available in the SISMER data base) and θ-S-σ (the HYDROCHANGES program that I initiated). For what concerns the Strait, seems to me that I performed quantitative analyses in e.g. Millot (2014a) or Sec. 3.2 in Part 2, but I do not see what kind of analysis could be qualified as quantitative in this Part 1. **Please, could you specify what is the quantitative analysis I make?**

*The profiles that originate from open-access databases (except for the data from Hydrochange sensors that are also invoked) are in themselves interesting and could provide some insights on the processes at plays.*
As for what concerns "*except for*", let me specify that I have always wanted to have the HYDROCHANGES data available to all participants to the program. And, for instance, you can note that my own data have been freely used by Garcia-Lafuente et al. (2015) even if I refused the offer I received for co-signing the paper (I disagree about too many points). I am sure anybody can ask my own team (I am now retired) and most colleagues taking part to the program for sharing their own data. That said, I note you find the profiles somehow "*interesting*".

*Most of the profiles (except for one section) are from the southern part of the western ALboran Sea in*

*the vicinity of the sill.*

- The first section (I call it transect) I analyze (Fig. 2) is at 6°05'W, in the western part of the Strait, just to demonstrate that the MO is far from being homogeneous there, contrary to what is claimed by Naranjo et al. (2015) and Garcia-Lafuente et al. (2017), the last two papers I know about the Strait.
- I then analyze (Fig. 6, 7) a north-south section/transect at 5°40'W, hence west of the Alboran (sorry to refuse using "Sea": **please, re-read carefully what I write in l. 135-140**), that is not in its southern part but across it, as clearly shown by Fig. 3.
- That being specified, it is right that **43** profiles are from the yo-yo time series in the "southern" (although relatively central, see Fig.3) part of the Alboran in the vicinity of the Camarinal southern sill while **8** and **6** profiles only are from the two transects. **Now, don't you think you should have mainly emphasized that a single 1-day yo-yo time series has provided more definitive information (and a larger amount of data) than two transects of similar duration?**

*A strong emphasis is given to the variety of properties (T, S, more than stratification) in different classes of potential density corresponding to a range of identified deep and intermediate source waters, modified by mixing either horizontally or vertically.*

- I disagree with "*T, S, more than stratification*". I first have to emphasize the fact that **"*stratification*" is misleading**. Just as an obvious example, the double-diffusivity process I mention (l. 988) is characterized by a stratification in both $\theta$ and S associated with a homogenization of the $\sigma$ profile! I suppose that you understand "*stratification*" as "density". And as I previously had to emphasize about your comment regarding "*T and S (and density)*", this is clearly wrong. **Please, add to the arguments presented above the fact that the series of $\theta$-S diagrams in Fig. 6, 8, and in Supplementary Information (SI) is presented together with $\sigma(z)$ profiles: my focus in mainly/only on "stratification in density"!**
- So as to try convincing you that **I am mainly interested in the stratification in density**, let me reformulate a comment I made about a question from Referee #1. I plotted the $\sigma(z)$ profiles with small dots, even when colored to identify this or that MW, in order to emphasize small scale features that are necessary to differentiate what I call "homogeneous layers" from what I call "interface layers": concretely, homogeneous (resp. interface) layers are associated with relatively "vertical" (resp. "inclined") portions of the $\sigma(z)$ profiles. And it is noteworthy that, within these overall vertical portions (relatively similar $\sigma$ values), one often notes marked irregularities associated with instabilities that evidence mixing processes. And I plotted the $\theta$-S diagrams with relatively large dots, in particular the colored ones, to emphasize the significance of the homogeneity of the colored waters as compared to the markedly changing characteristics of the waters in the interface layers.
- This being said, **don't you think you should have emphasized the fact that the "*different classes of potential density*" objectively defined by the distribution of the $\sigma_{max}$ values (Fig. 4, 5) were also representative of the distribution of all $\sigma$ values, as shown by both the blue curve in Fig. 4b, at least for the deepest/densest MWs and, for all MWs, by all $\theta$-S diagrams and $\sigma(z)$ profiles in Fig. 6, 8, SI ?**

*I have three main concerns: 1/ I have not really understood how the notion of a layer (in particular 'homogeneous' layer), or of an interface is defined ('by eye' or 'qualitatively' is mentioned). Of course*

*this is really difficult in this system with huge variability in mixing, stirring and internal tides, but some framework would be helpful to reach conclusions (and this is 'done' on figures 2, 6, 9 in some way for defining the layers from the vertical density profile).*

- I am sorry but "there is nothing to understand"! As you certainly noticed, and even though I am obviously aware about the difficulty in describing such a system, "I don't care" about *mixing, stirring and internal tides*, whatever their tremendous amplitude is, which has been demonstrated by numerous studies … including mine from the HYDROCHANGES data sets I and my team have collected at the Camarinal southern sill since 2003. I mean I want to be as objective/"naive" as possible and just try describing the σ(z) profiles I have, in particular from the yo-yo time series. And the image I objectively come with is that of steps with parts either relatively vertical (homogeneous layers) or horizontal/inclined (interface layers). The problem I thus had was to objectively separate both kind of layers, hence to define which parts of the plots should be associated with the homogeneous layers, hence colored and how. Doing this quantitatively, by specifying some kind of maximum dσ/dz gradient, was inadequate since some gradients within some (what I considered as) homogeneous layers were as large as gradients separating (what I considered as) homogeneous and interface layers. I do not want to speculate too much (i.e. trying to understand and explain) about the well-known processes you mention since I do not have any obvious arguments. And because I think that none of them can explain the observed features, I just put forward another hypothetical process possibly leading to such a multiple layering of the lower layer of MWs that is the overall north-south tilting of that layer at the Strait entrance (gravity making an angle with the density gradient that could be sufficient, even if only of a few degrees).

- What I did "qualitatively by eye" is: after having identified on the σ(z) profile a series of similar σ values over some depth range, which was very easy, I plotted them on the θ-S diagram, which first allowed me specifying the color (thanks to the separating isopycnals) and allowed me defining some cloud/patch. I then increased (decreased) the lower (upper) limits of the depth range step by step (dot by dot) and appreciated by eye from the θ-S diagram which dots could be considered as being a part of the cloud/patch or were out of that cloud/patch.

*2/ I got lost by the mix of detailed description of individual profiles (in particular thickness of different layers), without having a clear sense of what the author wants to demonstrate: is it quantifying the spread of properties (and distribution) in the different density ranges, and whether or not these are stratified or less-stratified layers separated with itnerfaces?*
I can obviously simplify the description of individual profiles and **I will strictly follow the Editor's recommendations**!
In any case, **I want to demonstrate that what I qualify as "homogeneous layers"**:

- are sometimes very thick, i.e. 164 m for the blue water at #47 of the time series (Fig. 9f), 83 m for the orange water at #5 of the 5°40'W transect (Fig. 6e). Note that even if the blue water is probably WDW, hence has always been identified at the Strait entrance, nobody has never imagined that WDW could have occupied about half of the depth range there. And note that even if the orange water is probably WIW, it is not yet recognized as separated from LIW (the red water) by, in particular, the two most recent papers about the Strait,

- are sometimes relatively thin and numerous, in particular the orange water in the south (#1, Fig. 6a), the center (#4, Fig. 6d) and the north (#6, Fig. 4f) of the 5°40'W transect. Horizontal sampling is clearly insufficient but, even if associated σ are different, it might be that these layers indicate some specific type of spatial distribution implying some specific process,

- often display marked density instability over large density ranges, i.e. density decreasing with

depth over several tens of m, as the violet water at #43 (Fig. 9e) or the red water at #6 (Fig. 6f),

- sometimes display very strange, i.e. never observed elsewhere in the Sea, distributions such as inversions in the slope of the θ-S diagram (the red water, #5, Fig.6e; the violet water, #9, Fig.9a) or a multiple layering (the red water, #21, Fig. 9b).
- necessarily imply the occurrence of interface layers in between (I probably did not catch your comment).

**I thus want to convince the reader that**, **in a so critical place** (at the entrance of the Strait), **so homogeneous layers** (density profiles in the remainder of the Sea are much more smooth) **having so specific characteristics** (instabilities or multiple layering are never observed in the remainder of the Sea) **necessarily reveal some specific and unforeseen processes of dramatic importance for addressing the controversy about homogeneity vs. heterogeneity of the MO. Are you convinced**?

*Is it to convince that these different water masses all take place in the overflow,*
Apart from the fact that I propose to reserve the term "overflow" for the DWs and the term "outflow" for the IWs, which justifies the Sec. 6.1.2 and 6.1.3 that you seemingly criticize and allows, I think, a correct understanding of the processes, **have you to be convinced that MWs so clearly identified at 5°43'W will "necessarily/very probably" flow out through the Camarinal sills (at 5°45'W, i.e. ~1,6 n.m. ~3 km to the west) as components of the MO**?

*and thus how?*
I am sorry to note that, asking *"how?"*, you just demonstrate that you did not carefully read the very last sentence of this Part 1 paper that is "Another deals with general circulation: how a set of horizontally stratified waters having to cross a narrow and shallow passage, with some waters outflowing and other overflowing, would organize in and after that passage, hence checking the validity of the schematic diagram proposed in Millot (2014a) and commented in Part 3?". **Maybe you can have a look at my 2014a paper … or read, and eventually accept reviewing, Part 3**?

*I suspect this last point might be tackled to some extent in the other two parts of the trilogy,*
Yes, as specified above, this point is "tackled" in Part 3. I mean that I can only present "what are just my own hypotheses", my aim being still to motivate theoretical analyses and numerical simulations that, I think, are necessary to validate data analyses.

*but it seems to me that some information on flow/transport is also required that is missing here.*
You know that published information about the (in- and out-) flow/transport rely, in particular, on specific S values of the MWs and the AWs assumed up to now that, I think, are not at all representative. Indeed, most of my papers about the Strait emphasize the heterogeneity of the MO (in particular S, see Fig.4,5) and Millot (2007) deals with the variability of the S(inflow). **I could just accept specifying that both flows are expected to be ~1Sv and differ by ~10%. Would you be satisfied with that**?

*If it is more the first objective, clearly more effort should be devoted to synthesize the results.*
I am sorry but I did not catch what was *"the first objective"*. In any case, **I will synthesize as much as required by the Editor**!

*To elaborate a little bit on that, I found figures 4 and 5 fairly synthetic ('the yoyo time series', although I missed there a plot of the layer thickness distributions).*

I am not sure I correctly understand your comment. I understand that i) you are satisfied with the synthetic character of Fig.4 and 5 (that are also fully objective, i.e. without any kind of personal analysis) and ii) you would have liked having a "*plot of the layer thickness distributions*", which is exactly done in Fig.8 (not so objective since based on my own definition of the layers). **Please, could you specify if you were actually satisfied by Fig. 4 and 5 and what you missed?**

*Actually, why is this presentation not also continued with the other two data sets?*

I am not sure I correctly understand your comment. I suppose that "*the other two data sets*" are those from the two transects (at 6°05'W and 5°40'W). If my understanding is right, my answer can only be "trivial". I mean that, while transects with such a relatively large sampling interval in space (hence at very different times) can only provide a set of different $\sigma_{max}$ values associated with each profile, only a yo-yo time series (same nominal location and sampling interval in time as short as possible, ~30' in this case) … or an HYDROCHANGES CTD ... can provide a well resolved time series of $\sigma_{max}$ and evidence periods of time with homogeneous characteristics, hence evidence homogeneous layers? **Did I correctly catch your comment?**

*On the other hand, figures 2,6, 9 which contain a wealth of information, are much too detailed and their presentation lost me.*

Having noticed that you recognize these figures "*contain a wealth of information*", I personally think that anybody's appreciation on the amount of details and the fact that one can appreciate the comments or be lost directly depends on the specific interest one can have in the specific processes evidenced by this new data set. I mean that if, apart from your own interest in the Sea and/or hydrology etc., you were directly interested by the MO characteristics, maybe you would have found the information not "too much detailed". In any case, **I will synthesize as much as required by the Editor!**

*Figure 7 could be considered a synthesis of figure 6. It is however a little difficult to interpret.*

Yes, Fig. 7 can be considered as a synthesis of Fig. 6 and it contains two types of information:

-the first information, and **I apologize I forgot to specify this**, presents an objective kriging of the data set. Fig.7a shows a sloping up southward of the MWs isopycnals, Fig. 7c and 7d show that MWs are cooler (warmer) and fresher (saltier) in the south (north).

-only the second information presents a synthesis of the coloring inferred from my definition of the various homogeneous layers as detailed in Fig.6. My own analysis of this synthetic information is clearly presented in l. 633-646 which ends with **"We let the reader open to any hypothesis for the time being" … so that I perfectly understand this information can be difficult to interpret … what I do (but you can have a different interpretation) in the Discussion.**

*Figure 8 summarizes the yoyo time series. If I understand it right, most of the deep layer is nearly unstratified (in density), with a very small portion of the water column in interfaces (this could be quantified).*

- **I am sorry but we have a totally different understanding of Fig.8**. While you think that

"*most of the deep layer is nearly unstratified (in density)*", I think that most of what you call "*the deep layer*", that is in fact the overall **lower** layer occupied by the MWs (be they qualified as intermediate -the IWs- or deep -the DWs-, the overall **upper** layer in green being occupied by the AWs) is in fact mostly composed of a set of homogeneous layers. A very small portion of this "*deep layer*" (my lower layer) of MWs being in the interfaces (in gray), **most of that "*deep/lower layer*" is dramatically stratified! Do you agree?**

- You can notice that I have reproduced in Fig.8 only the thickest of the interface layers, hence only for some profiles, while they should have been reproduced in between all homogeneous layers for all profiles, hence appearing as continuous undulating bands/layers from profiles #1 to #47 in the figure (that I did almost manually!). If you think that quantifying the homogeneous vs. interface layers thicknesses can be interesting and significantly inferred from my definitions, I can easily compute them … just by considering the sum of the various thicknesses (colored numbers) I specified in my figures with respect to the thickness of the whole lower layer (not colored in green in Fig.8). **Otherwise, I can just specify that no more than a few % of the lower layer of MWs is concerned by interface layers, hence that nearly the whole lower layer is stratified and composed of a set of homogeneous MWs. What do you recommend?**

*It would also be nice to mention how the tidal currents at this site evole in this deep layer (when is there eastward or westward flow?)*
The only reliable information I think I could provide (**I am not a specialist of that but maybe you could help me?**), would be a plot of the theoretical sea-surface tidal signal (mainly the barotropic tide) at some nearby location. Now I know, initially from the large amount of papers presenting in situ CTD profiles, but also from my own HYDROCHANGES data at the Camarinal southern sill as well as from satellite data about the internal bore, that the internal (baroclinic) tide has a tremendous amplitude there. This being said, I am sorry but I do not know how reliable would be an information about the associated baroclinic tidal current … and, considering the large scale (MO) current (known to be relatively intense), … what would be the resultant effective current (furthermore I demonstrate the MO is markedly layered in density, hence necessarily in velocity!). And just because internal / baroclinic features are more variable / less predictable than surface / barotropic ones, what would be the significance of any comparison with a 23-h (less than 2 semi-diurnal periods) time series? **In other words, I do not share your opinion about the direct relationship between "*tidal currents*" and "*when is there eastward or westward flow?*" that is in fact "tidal+long time scale current or mean current". Please, could you either argue in answering my own comments and help me with adequate references or just agree with me?**

*3/ the structure and contents are not what is expected of a scientific paper. There are lots of judging comments on other published work or general statements which dont have their place in the manuscript (starting in the introduction).*
**This comment is for me a very serious accusation that I need denying by a listing and justification of all my "*judging comments on other published work or general statements*".**

**As for the Introduction:**
- l. 76-80: "Even though the MO has been intensively studied from the point of view of strait dynamics (maximal vs. sub-maximal regimes, inflowing vs. outflowing amounts, tidal internal waves and currents, bottom friction and associated turbulent processes, etc.), very few studies have focused on its hydrological characteristics … A large percentage of old (before 2000)

papers have considered that the spatial and temporal differences in the MO hydrological characteristics within the Strait were nothing else than natural variability and have postulated the overall homogeneity of the MO from the Strait entrance (near 5°45'W)... ". Obviously, I know that all (sic) experimental studies dealing with in situ data in the Strait surroundings have presented CTD data, hence have discussed hydrological characteristics, but **I maintain that, at least to my knowledge:**

- **i) most (not to say all) have only addressed strait dynamics (see the list I make),**
- **ii) very few (not to say none) have focused on the MO (sensu stricto) characteristics addressing its homogeneity vs. heterogeneity,**
- **iii) a large percentage (not to say all) have postulated the overall homogeneity of the MO.**

**Please, could you give me at least one reference specifically addressing the homogeneity vs. heterogeneity of the MO?**

- l. 91-93: "... the most recent papers (2017) now hypothesize that the MO is heterogeneous at the Strait entrance before becoming homogeneous within the Strait itself and then being split into veins just from the Strait exit." **I obviously maintain my writing** and specify that, because the team considered as the specialist of the Strait entrance has dramatically changed its point of view very recently (in 2015):
  - i) it is not necessary to refer to the older papers assuming a homogeneity at the Strait entrance (hence, in particular, to all other papers from that team),
  - ii) my Fig.2 definitively demonstrates the MO is heterogeneous within the Strait itself (at 6°05'W in this case) and is dedicated to this assertion.

**Do you agree that my l. 91-93 correctly / honestly report the hypothesis now formulated by Garcia-Lafuente et al. (2017) without any judgment and, please, could you specify what is the previous paper, if any, of these authors assuming such an hypothesis?**
**And what is the hypothesis these authors have previously formulated?**

- l. 96-97: "The main reason is that the postulate about the MO homogeneity has evolved since, initially about the MO from the Strait entrance, it is now, thanks to our papers, only within the Strait itself ...". **I do think that:**
  - i) the homogeneity of the MO at the Strait entrance has always been a postulate (sic, i.e. without any specific analysis such as the one I present),
  - ii) the changes from "homogeneity at the Strait entrance" to "homogeneity within the Strait itself" are only due to my own previous papers.

**Please, consider the 3rd paragraph of p. 42 of Naranjo et al. (2015) that clearly specifies the opposition between Millot (2014a) and the "widespread view expressed by the Baringer et al. relatively old papers", as well as the agreement recognized with my own point of view in the 1st paragraph of the 5.1 section of this paper, and could you let me know what is now your own thinking?**

- l. 99-100: "... very few old papers have noticed incoherency between such a process and some of their own data without providing sound explanations; ...". The only papers I know are the Howe's ones and, even though I "admire" the analysis Howe made and the position he took in spite of everyone, I think he could not provide the explanations I am now able to provide. **My comment is thus positive and worth being specified here since I need to say I am not the only one rejecting or having rejected the MO homogeneity postulate at the Strait entrance. Do you agree?**
- l. 102-103: "... just because neither a single in situ experiment nor a single theoretical analysis have ever been dedicated to specify the homogeneity vs. heterogeneity of the MO within the

Strait,...". **In case you do not agree, please could you specify which experiment or simulation tackled the homogeneity vs. heterogeneity question?** **And don't you think this is a comment worth to be specified?**

- l. 126: "We finally discuss (Sect. 6) some generally forgotten evidences...". I do think that all (sic) the evidences I mention (all MWs must exit, the IWs outflow, the DWs overflow, generic acronyms must be given, the eastern DWs must be identified, SAW and NACW must be differentiated) are not considered in any of the published papers, in particular the last one that is Garcia-Lafuente et al. (2017). **Don't you think this must be specified?**

**As for the Background:**
- l. 137-138: Don't you think that **you would never have accepted from a student** a writing using both "the Alboran Sea" and the "Alboran Basin" in a single abstract or in any text**?**
- l. 147-148: Don't you think **it is necessary to inform the reader** that the significance of a given acronym (WIW) has been officially changed since a long time (2001)**?**
- l. 149-151: Don't you think it is necessary to note that a given acronym (LIW) has been **improperly used (also by myself till 2014!**) and continue to be improperly used**?**
- l. 152-156: Don't you think **it is necessary to specify the true paternity** of some given statement or hypothesis at least (about TDW)**?**
- l. 164-175: Don't you think **it is necessary to specify that the team who published the last two papers about the Strait and is considered as the specialist of the Strait entrance:**
  - i) has recognized, up to 2015, only two MWs,
  - ii) continues claiming that WIW is embedded within LIW, contrary to what is demonstrated since 1985 and herein,
  - iii) misreported about TDW/EDW and totally ignores my comment (6.1.1) about the necessity for all MWs to outflow,
  - iv) invoke a process (a Bernoulli suction) that, I think, is absolutely not necessary**?**
- l. 197-205: Don't you think **it is necessary to reproduce (so as to avoid any judgment by myself!**) sentences from the two last papers about the Strait that explain why the MO homogeneity is so largely assumed**?**
- l. 206-208: Don't you think **it is necessary to specify that most (not to say all) recent papers** (I will add Naranjo et al. (2015) in the list, see my comment about l. 96-97) **do not detail the MO characteristics and refer to the old Baringer and Price's papers?**

In case you think I express ***"judging comments on other published work or general statements"*** in the remainder of my paper, **please let me know!**

*The presentation starts with an interesting figure 1, which is somehwat followed during 2, but not really afterwards (so I wonder how important it is for the paper), until mentioned again in section 6.1.6.*
I appreciate the interest you found in Fig.1 but I am not sure about what you mean with "*which is somehwat followed during 2*". I suppose you mean that Fig. 2 interestingly follows Fig.1 and I suppose you noticed that Fig.1 is a conceptual representation of the processes leading to features such as those evidenced by the data at 6°05'W in Fig.2. **Fig.1 and Fig.2 have a dramatic importance for the Strait dynamics understanding in general and my trilogy in particular, and this importance is emphasized, as soon as the Introduction, with three sentences**:
- l. 89-93: "Even though the MO characteristics and dynamics are driven by processes that are

different from place to place, this controversy is an overall problem that cannot be truncated, furthermore the most recent papers (2017) now hypothesize that the MO is heterogeneous at the Strait entrance before becoming homogeneous within the Strait itself and then being split into veins just from the Strait exit." The most recent paper (up to now) I think about is Garcia-Lafuente et al. (2017) and **I think I honestly report about this paper.**

- l. 106-109: "Now, presenting our arguments in such a selection would have provided the reader with a complete but tremendously complicated overview that, we think, is unnecessary since we are now able to clearly demonstrate, with the analysis of a single θ-S diagram (Sect. 3), that the MO is definitively heterogeneous within the Strait itself (at 6°05'W and upstream)". The single θ-S diagram in Sect. 3 is presented in Fig.2 and analyzed on the basis of Fig.1; it clearly demonstrates that **the postulate about the MO homogeneity**, initially (up to ~2015) at the Strait entrance and now (early 2017) within the Strait, **is wrong**.

- l. 109-111: "With such a basic evidence in mind, the reader is thus proposed, all along our trilogy, to make his/her own point of view about the characteristics of the MO heterogeneity, our personal results and analyses being only proposed as guidelines in the various Discussion sections." **specifies my own attitude all along my trilogy:**

**Having demonstrated with Fig.1 and Fig.2 that the MO is heterogeneous within the Strait, hence at the Strait entrance, I let the reader making her/his own analysis of all the data sets I present.**

*The presentation of individual profiles is spred over many pages and with figures 2, 6 and 9. I am wondering whether they could not be summarized in a joint section.*
**I suppose you have noticed that**:

- Fig.2 presents profiles along a **transect** at 6°05'W, Figs.6 present profiles along a **transect** at 5°40'W and Figs.9 present profiles from a yo-yo **time series at the specific location** 5°45'W-35°55'N.

- profiles along **the transect** across the Alboran, even if they sampled the deepest part of this sub-basin, **do not evidence** the blue/densest MW (Figs.6), **contrary to the time series** at a relatively shallow location and intermediate latitude (Figs.9), this blue/densest MW being retrieved, even if significantly mixed, at #4 (Fig.2). Also, for instance, layering of the lower layer of MWs within the Sea that dramatically increases from 5°40'W to 5°45'W totally disappears at 6°05'W.

Considering such features are very specific from place to place, I do not see the interest of having "*their presentation summarized in a joint section*" and **I think that so specific features can be more interestingly compared in the Discussion section.**

*Then, the discussion starts with a very long 6.1 entitled: 'Some generally forgotten evidences'. Frankly, I dont think that the assumption should be that we 'readers' have forgotten some rather general considerations on the western Med ocean circulation and processes taking place near Gibraltar Strait. As I stated, maybe 6.1.6 could be summarized and kept.*
I am pleased to note that you, as a referee, don't have forgotten some rather general considerations about the Sea functioning. Therefore, **please have a look at Garcia-Lafuente et al. (2017)**, have in mind that they are considered as the specialists (of the Strait entrance) I am "fighting with", **and check that they**:

- never list the MWs and specify where/how they exit from the Sea,
- do not make any difference between outflowing and overflowing MWs (my IWs and DWs),

- do not give generic acronyms (they still deal with LIW),
- never mention the DWs from the eastern basin,
- never differentiate SAW from NACW.

**Don't you think I have to detail all the critics I address to the team actually considered as the specialist of the Strait entrance functioning?**

*6.3 deals with what should be done next. I dont think that this has a place in a discussion section. This discussion of what could be done next is indeed continued in part of the conclusion section, which might not be the right place for it.*

I am sorry but your last sentence is not very clear for me since it can be interpreted as "the conclusion section might not be the right place". In any case, I personally think that:

- the data presented in Fig.4a and 5 display **raw features never evidenced up to now,**
- they were collected using **a strategy rarely used in the Strait** in particular,
- they were **collected at a very specific place**, just east (~3 km) from the Camarinal sills,
- they reveal that **MWs are involved in strange processes never imagined before.**

So that I think **it is worth:**

- **i) detailing** why collecting yo-yo time series might be more efficient than performing transects,
- **ii) proposing** working hypotheses about possible processes and adequate sampling strategy.

Furthermore "*next*" means "in the Conclusion", **I thus continue thinking that it is worth having such details and arguments in a Discussion sub-section, with only a summary of them in the Conclusion. But I will follow the Editors recommendations!**

*Alltogether, I feel that the paper should be strongly streamlined (at least, cut by half),*

**I will follow the Editors recommendations!**

*and edited to avoid statements that can be interepreted as judgmental or oversimplifying, or not directly related to the work presented.*

**I refute the comment that I should have presented "*judgmental or oversimplifying or not-directly-related-to-my-work statements*". Could you provide me with a list, or at list with some examples of such statements?**

*Some of the profile figures could be kept in supplementary material, with a short descriptive presentation of a couple of typical examples and more effort to dscribe synthetically the variety of profile characteristics.*

As long as "*supplementary material*" is seemingly nothing else than a pdf file in which I can apparently present **figures and comments**, I have no problems with reducing the numbers of figures in the paper. **But, if I understand you correctly, you ask me to "*dscribe synthetically the variety of profile characteristics*" with only "*a couple of typical examples*" … hence saying to the reader "believe me, you do not have to appreciate my comments by yourself". Is it what you are suggesting?**

*One question: the ranges of density given for the different watermasses is rather narrow: how is it taking care of the low frequency (multidecadal) evolution of deep and intermediate Mediterranean*

*water masses*

One answer in several points:

- I am the first presenting and analyzing, in this Part 1 paper, 1985-1986 data available for free to anybody since 2002 that demonstrate a marked and unexpected layering, within relatively narrow density ranges, of the MO at the Strait entrance,
- I, alone at the beginning, proposed the strategy of deploying autonomous CTDs to monitor **low-frequency changes** in the Sea, which lead to the HYDROCHANGES program I thus initiated,
- I am the first having demonstrated, during the 2003 CIESM Congress in Barcelona and with Millot et al. (2006), that the MO was encountering marked **low-frequency changes** at the Strait entrance.
- If you have time and/or willingness to have a look at my Part 3 paper, even though just at Sect. 4.8 and Fig.20, you will see that I am the first demonstrating that the whole MO, from the Strait entrance to the Strait exit, has encountered **low-frequency changes** in both its hydrological and dynamical characteristics,
- I am also the first having collected and published (Millot, 2007) data demonstrating that the MI (the Mediterranean Inflow of AWs) was also encountering **low-frequency changes**.

I suppose you now agree that **I am tremendously interested by the low-frequency changes.** That being said, **I essentially want "to demonstrate",** and "**I don't take care about speculating with definitive types of observations collected in different decades"** (in the mid 1980's for the layering and from the mid 1980's to the late 2000's for the long-term changes). **I imagine and can just hypothesize that the actual layering is roughly similar to the mid 1980's one, just being shifted in the θ-S-σ ranges.**

**To conclude**, **this hypothesis … and all my other hypotheses as well** **(made in the Part 2 and Part 3 papers) …** **can be easily checked** **with either a single few-h long yo-yo time series (at the Strait entrance) or a couple of ~1-day back and forth cross-Strait transects (within the Strait and at the Strait exit).**

**Therefore, please, consider that:**

**i)** **recommending the publication will motivate additional studies** **(data collections, theoretical analyses, numerical simulations)** **that will necessarily either definitively discredit me or provide the whole community with a new and promising point of view!**

**ii) being retired, hence no more "obsessed" by the length of my CV, still passionate and in as good health as possible, having often been in opposition with the "general thinking" and furthermore fully convinced I am right,** **refusing the publication will not prevent me from submitting (and finally publishing) in any other journal****, whatever the notoriety of this journal would be and whenever this would be possible!**

---

## Author Comment (AC3) · 14 Nov 2017

To the Editor, the referees and all readers of my paper(s),

I would like to take advantage of the opportunity offerred by OS to post additional comments that will allow the review process to continue.

I think that both referees did not (enough) emphasize the fact that the main interest of my paper(s) is not in my own analyzes, furthermore I underlined my overall attitude (l. 109-111) with "... the reader is thus proposed, all along our trilogy, to make his/her own point of view about the characteristics of the MO heterogeneity, our personal results and analyzes being only proposed as guidelines ...".

As for this Part 1 paper, and on the basis of my relatively long (I am now retired after more than 45 years spent to publish more than 100 papers) experience, I think I very rarely had the chance to demonstrate something with only a set of raw data and without any need for either theoretical analyzes or numerical simulations. And I had the chance twice with the data sets presented in Fig.2 and Fig.4a that can be considered without any color or additional information.

Figure 2 demonstrates (sic) that the MO is heterogeneous at 6°05'W, that is in the western part of the Strait, which contradicts the assertions expressed in the most recent papers published about the Strait (Naranjo et al. (2015) and Garcia-Lafuente et al. (2017)).

Figure 4a demonstrates (sic) that the MO is heterogeneous just ~3km upstream from the Camarinal sills, being composed there of a set of homogeneous layers obviously associated with a set of generic MWs that can be identified only in density (neither in temperature nor salinity), hence contradicting what has been assumed up to now about the MWs identification.

Finally, I think that both referees did not (enough) emphasize the interest of performing yo-yo CTD time series at the Strait entrance in particular, furthermore this can be done in a relatively easy way, and did not (enough) stress the necessity of performing dedicated theoretical analyzes and numerical simulations.